# Investigating the Effects of Microclimate on Physiological Stress and Brain Function with Data Science and Wearables

**Kenneth Y. T. Lim** [1,*] **, Minh Anh Nguyen Duc** [2] **, Minh Tuan Nguyen Thien** [2] **, Rajamanickam Yuvaraj** [1] **and Jack S. Fogarty** [1]

[1] National Institute of Education, Nanyang Technological University, Singapore 637616, Singapore
[2] Independent Researcher, Singapore 357689, Singapore
* Correspondence: kenneth.lim@nie.edu.sg

**Abstract:** This paper reports a study conducted by students as an independent research project under the mentorship of a research scientist at the National Institute of Education, Singapore. The aim of the study was to explore the relationships between local environmental stressors and physiological responses from the perspective of citizen science. Starting from July 2021, data from EEG headsets were complemented by those obtained from smartwatches (namely heart rate and its variability and body temperature and stress score). Identical units of a wearable device containing environmental sensors (such as ambient temperature, air pressure, infrared radiation, and relative humidity) were designed and worn, respectively, by five adolescents for the same period. More than 100,000 data points of different types—neurological, physiological, and environmental—were eventually collected and were processed through a random forest regression model and deep learning models. The results showed that the most influential microclimatic factors on the biometric indicators were noise and the concentrations of carbon dioxide and dust. Subsequently, more complex inferences were made from the Shapley value interpretation of the regression models. Such findings suggest implications for the design of living conditions with respect to the interaction of the microclimate and human health and comfort.

**Keywords:** environmental data; microclimate; electroencephalography (EEG); physical health; mental health; sensors; machine learning; internet of things; citizen science

## 1. Introduction

Climate change has been one of the most urgent problems to confront in the 21st century. The fifth report of the Intergovernmental Panel on Climate Change (IPCC) confirms that the human influence on the climate system is clear and growing, with impacts observed across all continents and oceans [1]. Climate change has been degrading the quality of life for every creature on Earth. Glaciers have shrunk, ice on rivers and lakes is breaking up earlier, plant and animal ranges have shifted, and trees are flowering sooner [2].

From the United Nations World Urbanization Prospects in 2017, 4.1 billion people were living in urban areas [3]. This means over half of the world (56% in 2020) lives in urban settings. Urbanization continues apace, and it also accounts for global climate change. The rapid and large-scale urbanization leads to severe land-use conversion and impacts ecosystem services [4]. The latter refer to the direct and indirect contributions of ecosystems to human well-being. With ecosystems being damaged, human well-being is also affected. In the context of this pressing problem, urban microclimate studies have been gaining prominence due to rapid urbanization [5].

A microclimate is a small area within a surrounding larger area with a different climate [6]. Any given climatic region therefore comprises many other types of microclimates, which vary in characteristics from the region as a whole. Because our planet in general is broadly conducive to life, we—as humans—have populated its land masses. Comparing

the human scale to that of the various habitats in which we live, the difference of these scales means that changes in the climates of these habitats may disproportionately affect the conduct of our daily activities.

Concurrently, with the rapid changes in microclimate, a technological revolution has also brought fresh wind to the field of neuroscience. Collecting electroencephalogram (EEG) data has been made progressively more accessible and that has paved the way for many researchers to delve into the activities of the brain. For example, the EEG is used in photic stimulation, a common procedure performed in the EEG laboratory in children and adults to detect abnormal epileptogenic sensitivity to flickering light (i.e., photosensitivity) [7]. In another instance, sound level can have an impact on the functioning of the brain and can be observed through EEG data [8].

In research, the EEG is used as a means of identifying human stress level as it has already been proved that there is a significant correlation between levels of psychological stress and EEG power [9]. In other cases, EEG data are also strongly related to the changes in human physiological health, such as in heart rate variability [10].

That is the reason why the authors felt compelled to explore the relationships between microclimate and our brain activities. It is self-evident that climate change has various effects on the well-being of a person. As human beings, we are conscious and aware of our surroundings, and our responses to changes in the microclimate may affect our emotions and health. To elaborate, climate change might precipitate changes to microclimates to the extent that for those inhabiting these biomes the changes might be detrimental to physical and mental well-being. For instance, a study by Liu et al. in 2019 concluded that "the increasing research interest in thermal comfort and health has heightened the need to figure out how the human body responds, both psychologically and physiologically, to different microclimates" [11]. Therefore, investigating EEG data may unveil hidden relationships as to how microclimate is related to our perception of well-being at a granular level.

This paper reports a study conducted by students as an independent research project under the mentorship of a research scientist at the National Institute of Education, Singapore. DIY EEG headsets were designed and built from a citizen science approach to collect brainwave data daily. The data from these sensors were complemented by those obtained from smartwatches. As a third source of data, wearable devices containing environmental sensors were designed and worn, respectively, by five adolescents from July 2021 to June 2022.

Among other studies, our work was inspired by the earlier work of Palme and Salvati [12], in which they concluded that modified urban microclimates had a deep impact on the comfort of the inhabitants. Palme and Salvati lamented that while there have been various studies conducted on the effects of urbanization on the microclimate and on how the microclimate has changed human health in the efforts to redesign and restructure urban areas, there have been relatively few studies on the relationships between the microclimate and human health and emotions [12].

The aim of the study was to explore the relationships between local environmental stressors and physiological responses from the perspective of citizen science. The authors were interested in understanding the extent to which the data from DIY EEG headsets would be comparable to that from industrial grade headsets, with a view to democratizing the collection of such data for the investigation of these relationships. From our review of literature which follows, we know that microclimate affects physiological and mental well-being; hence, we were interested to understand the relationships between microclimate and brain activity, as well as how brainwaves affect our physiological and mental well-being. We see one of the potential contributions of our work to be our use of self-designed, low-cost, wearable units for measuring microclimate, as well as EEG headsets. The relatively low cost of these wearables has positive implications for the affordance of the scalability and—consequently—on crowd-sourced citizen science in this as yet under-reported field of the relationships between microclimate and well-being.

## 2. Review of Literature

### 2.1. The Importance of Microclimate

Microclimate has been defined as the suite of climatic conditions measured in localized areas near the earth's surface [13]. Microclimate includes environmental variables, such as temperature, light, wind speed, and moisture. It has been critical throughout human history, providing meaningful indicators for habitat selection and other activities [14]. Regardless of the global biomes in which we live, it is specifically microclimate that our bodies respond to, and not to the descriptors of the respective climatic region as a whole. For example, farmers have long used localized seasonal changes in temperature and precipitation to schedule their agricultural activities. Microclimate directly influences ecological processes and reflects subtle changes in ecosystem function and landscape structure across geographical scales [15].

On the spectrum of well-being, brainwaves can be an indication of physical and mental well-being and productivity as they influence several EEG rhythmic frequencies [16]. This has been evident in the use of the EEG to detect abnormal epileptogenic sensitivity to flickering light (i.e., photosensitivity) [7] and how, for instance, different sound levels have different effects on the functioning of the brain, which can be observed through EEG data [8]. Specifically, with the rise in noise levels, the relative power of the alpha band increases while the relative power of the beta band decreases as compared to background noise. The most prominent change in the relative power of the alpha and beta bands occurs in the occipital and frontal regions of the brain, respectively.

It can therefore be seen that microclimate weighs heavily in the effect on physiological and mental well-being; hence, there should also be a relationship between microclimate and how it affects brainwaves, as well as how brainwaves affect our physiological and mental well-being.

As an example related to human health, microclimate in urban areas affects our thermal comfort [12]. However, the relationships between microclimate and biological processes are complex and often nonlinear. For example, plant distribution can be perceived as a function of light, temperature, moisture, and vapor deficit [14]. Therefore, just a subtle change in microclimate could cause detrimental effects on human emotions and health, aside from just thermal comfort.

### 2.2. Effects of Urbanization on Microclimate

Rapid urbanization, especially in developing countries, has led to large flows of migration to urban areas [16]. According to the market research firm Statista, the degree of urbanization worldwide was at around 56% in 2020 [17]. With rapid urbanization, changes to the urban environment and climate are inevitable [18]. The rate of urbanization is very high, and the anthropogenic effects on Earth's climate are difficult to predict.

At local scales, activities associated with land use and land cover changes and urbanization induce impacts such as changes in atmospheric composition in water and energy balances and changes in the ecosystem [19]. By definition, ecosystems are interconnected; therefore, a small change in any component can result in non-linear effects elsewhere. For example, according to a study conducted by Xiong et al. in 2015 on the influence of different air temperature step changes on human health and thermal comfort, perspiration, eye-strain, dizziness, accelerated respiration, and heart rate were all sensitive self-reported symptoms [20].

Due to global climate change and intensifying urban heat island effects, urban living environments have deteriorated, becoming increasingly detrimental to human thermal comfort and health, not only psychologically, such as in terms of thermal sensation, mood, and concentration, but also physiologically by way of, for example, sunburn, heat stroke, and heat cramps [11]. They also cautioned that "global climate change and intensifying heat islands have reduced human thermal comfort and health in urban outdoor environments". Other studies on renaturing cities have found that changes to urban microclimate can potentially exacerbate the risks of meteorological hazards such as heatwaves. Heat-related

issues not only have an impact on the environment but can also lead to heat-related human health problems and, in extreme cases, cause deaths [19]. This is only one aspect of how the microclimate can affect humans.

### 2.3. Implications of Electroencephalographic (EEG) Data

With reference to the preceding discussion on the importance of microclimate, it is hypothesized that different states of the surrounding environment can have different effects on humans and, as such, EEG signals are used as a means of identifying human stress levels as it has already been proved that there is a significant correlation between levels of psychological stress and EEG power [9]. In other cases, EEG data are also strongly related to the changes in human physiological health, such as in heart rate variability [10].

With regard to the technicalities of the different brainwaves, the gamma, beta, alpha, theta, and delta waves correspond to the frequencies of 0–4 Hz, 4–7 Hz, 8–12 Hz, 12–30 Hz, and 30–100+ Hz, respectively.

Brain activities around the frontal, temporal, and occipital lobes are commonly collected, analyzed, and used to detect fatigue and attentiveness [21]. EEG signals can be analyzed in the time domain, as in the case of event-related potentials (ERP) [22], as well as in the frequency domain, as with the spectral content of frequency bands [23]. More recent EEG analysis methods involve the use of functional connectivity [24] and source separation [25] algorithms. Signal source separation methods for EEG signal analysis, such as the Moore–Penrose pseudoinversion, as proposed in [25], allow the solving of the inverse problem of neural recordings and the spatial identification of the different sources in the brain responsible for specific neural activations. Recently, inter-subject correlation (ISC) of the EEG has been proposed as a marker of attentional engagement [26–28]. With the aforementioned implications of different brainwaves, EEG data can be sensibly used as a means of investigating human productivity [29].

## 3. Materials and Methods

As stated in a preceding section, this paper reports a study conducted by students as an independent research project under the mentorship of a research scientist at the National Institute of Education, Singapore. DIY EEG headsets were designed and built from a citizen science approach to collect brainwave data daily for two-hour sessions each day. From July 2021 to June 2022, the data from these sensors were complemented by those obtained from smartwatches (namely blood oxygen saturation, heart rate and its variability, body temperature, respiration rate, and sleep score). As a third source of data, identical units of a wearable device containing environmental sensors (such as ambient temperature, air pressure, infrared radiation, and relative humidity) were designed and worn, respectively, by five adolescents over the same period.

### 3.1. Collecting Microclimate Data

To collect the microclimate data, a small portable device that could be worn on the waist was designed in order to measure the following ambient environmental conditions:

- Noise level;
- Infrared radiation through light intensity;
- The amount of dust;
- Carbon dioxide concentration;
- Temperature;
- Relative humidity;
- Air pressure.

In total, five such devices were built by the authorial team. Each device was worn by the first and second authors and by their adolescent peers. The devices measured 12 cm by 6 cm by 2 cm, respectively, and contained a low-cost sensor for each of the preceding environmental variables listed, together with a battery with sufficient capacity to power

the sensors over the course of a typical day. The device could be secured to a belt by two regular clothes clips.

Every five minutes, each device would automatically log its measurements onto a designated cloud-based spreadsheet. At the same time, the device would ping the nearest publicly accessible weather station (as provided by the Singapore Meteorological Service) for the wind speed and wind direction prevailing at that time. Figures 1 and 2 show schematics of the components, and Figure 3 shows an assembled device.

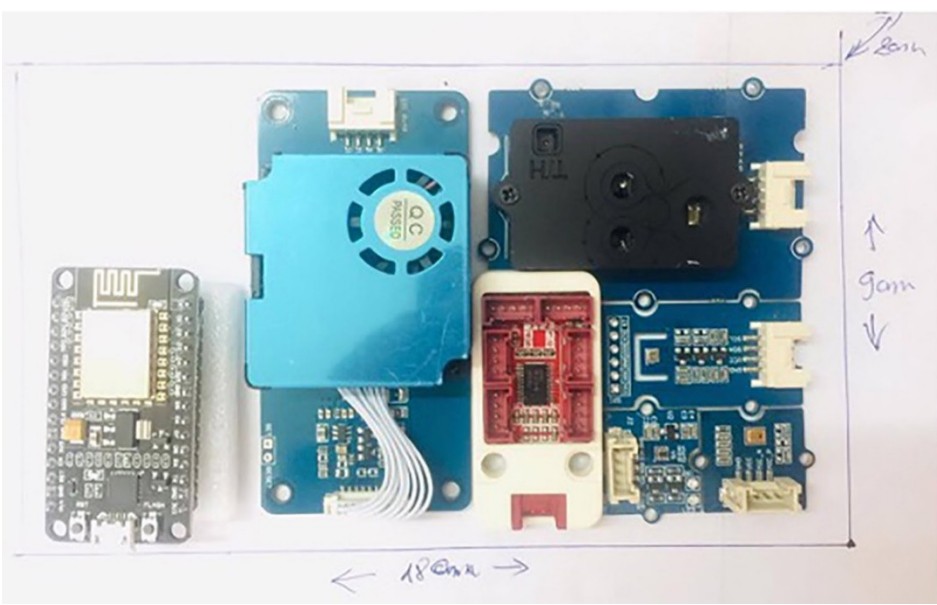

**Figure 1.** Early schematic of sensors for the wearable.

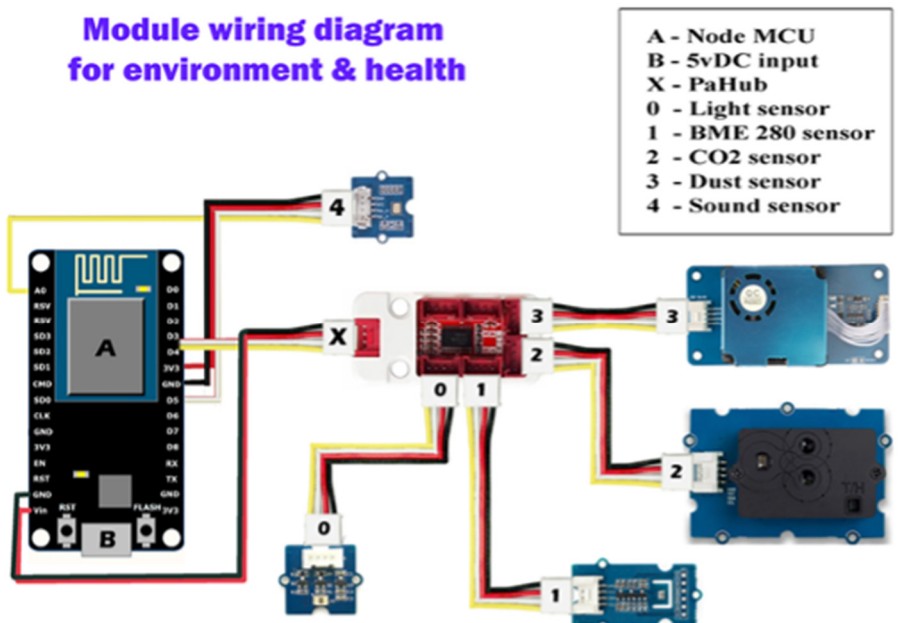

**Figure 2.** Late schematic of sensors for the wearable.

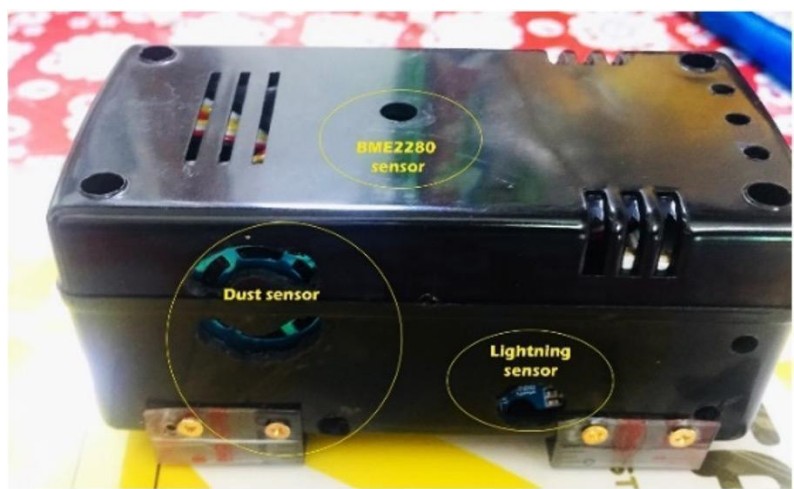

**Figure 3.** Assembled wearable device.

*3.2. Collecting Data on Mental State and Health*

Biometric data were collected using three Huawei Honor Band 6 smartwatches and two Fitbit Sense smartwatches. The factors measured and recorded are:

- heart rate.

This variable, in combination with the "Oxygen saturation" measured, was used by both types of watches to generate a so-called stress score (the latter was pertinent to waking moments and arbitrarily indexed from 1 to 100).

In addition, the Fitbit Sense watches were able to measure:

- skin temperature at the wrist (indicated as degrees Celsius off the baseline of the core body temperature);
- respiration rate (measured in breaths per minute);
- heart rate variability, which is the physiological phenomenon of variation in the time interval between heartbeats.

Heart rate variability was—in turn—indicated through the variables of:

- root mean square of successive differences between successive beat-to-beat intervals (rMSSD).

In earlier work by Izard [30], it was recognized that there was a high degree of connectivity among the neural structures of the brain and its systems. Emotions and cognition, although having separate features and influences, are dialectic, integrated, and co-mingled in the brain. It has been suggested that emotions play a central role in the evolution of consciousness, influence the emergence of higher levels of awareness, and largely determine the content and focus of consciousness throughout one's lifespan [30]. As such, by attempting to explore the connections between the participants' stress scores, brain activities, and the corresponding biometric data, we can perhaps glimpse how the participants respond physiologically and affectively when they are in a specific microclimate.

*3.3. Collecting Encephalographic (EEG) Data*

As this research focuses on citizen science, the expenses derived from the procedure should be minimal. With that in mind, a DIY EEG headset was self-designed and built and customized to each participant using common household items, namely sponges and tapes. In addition to the built frame of the headset, IDUN Dryodes were used as sensors to measure and collect EEG data.

Each brain hemisphere (the parts of the cerebrum) has four sections, called lobes. Located in front of the head, the frontal lobe is the largest lobe of the brain. The frontal lobe is responsible for personality characteristics, decision making, movement, and processing smell, as well as speech ability. In this research, channels Fp1 and Fp2 are located at the

frontal lobe. As for the parietal lobe, it is located at the middle part of the brain and is responsible for identifying objects and understanding spatial relationships. The parietal lobe is also involved in interpreting pain and touch in the body. The parietal lobe also helps the brain understand spoken language. In this research, channels C3 and C4 are located at the parietal lobe. As for the occipital lobe, it is located at the back of the brain, and it is mainly responsible for vision. In this research, channels O1 and O2 are located at the occipital lobe. Finally, located at the side of the brain, the temporal lobes are responsible for short-term memory, speech, musical rhythm, and some degree of smell recognition. In this research, channels T5 and T6 are located at the temporal lobe.

The EEG headsets are customized to each participant's head size, and each comes with 10 channels to collect data: Fp1—Fp2—C3—C4—T5—T6—O1—O2 and 2 reference dryodes (10–20 system), where Fp is an abbreviation for the frontal lobe, C is for the central part of the brain (parietal lobe), T is for the temporal lobe, and O is for the occipital lobe.

### 3.4. Procedure for Validation of EEG Data Collected by DIY EEG Headset

In total, there are nine segments in which the participants' brain signals are recorded in different environmental conditions. The participants were each assigned a room, in which they were seated, in order to complete a geographically themed pen-and-paper quiz within an hour. Each participant went through two sessions, first with a DIY EEG headset and then with an ANT Neuro EEG headset. During the intervals between the different actions, the participants were also required to fill in a ten-point Likert scale in terms of how they felt mentally and physically. The results were to be compared and classified into segments with machine learning models to determine whether the changes in environmental conditions affected the EEG signals. The channels in the recorded DIY EEG were Fp1—Fp2—C3—C4—T5—T6—O1—O2 (10–20 system), and the equivalent channels in the ANT Neuro EEG headset were Fp1—Fp2—C3—C4—P7—P8—O1—O2 (10-10 system). Figure 4 presents the standard of procedure.

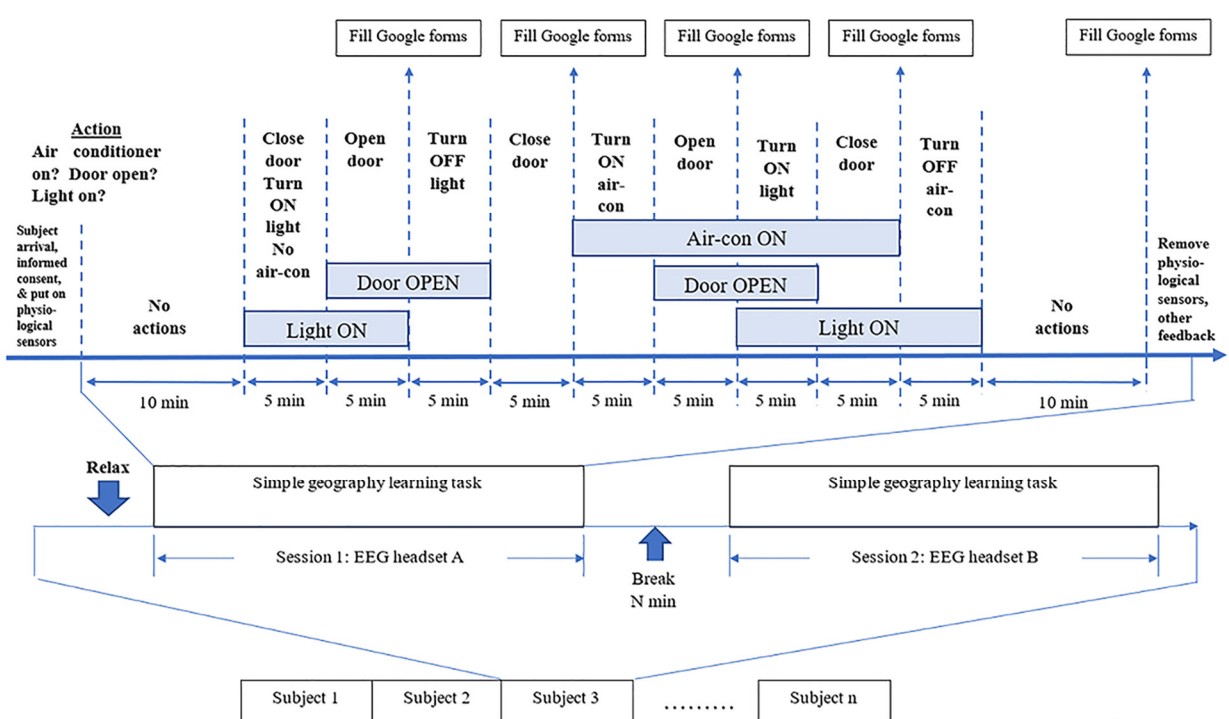

**Figure 4.** Standard of procedure of validation of DIY EEG data collected.

It was concluded that the DIY headset's EEG data points were in range when compared with the research grade equipment. Following that, whether there was a significant

difference in the signals between the segments was determined or, in other words, whether or not the environmental factors affected the EEG recordings with four-second epochs. Table 1 shares the results of the ANOVA test that was performed with the non-linear features of the signals, namely skewness, detrended fluctuation analysis, approximate entropy, and sample entropy.

**Table 1.** Results of ANOVA test between research-grade equipment and DIY EEG headset.

|  | *p*-Value | |
|---|---|---|
|  | **ANT Neuro** | **DIY** |
| Skewness | $5.56 \times 10^{-49}$ | $9.39 \times 10^{-81}$ |
| DFA | $7.54 \times 10^{-89}$ | $6.3 \times 10^{-102}$ |
| Approx. entropy | $3.31 \times 10^{-38}$ | $3.24 \times 10^{-66}$ |
| Sample entropy | $4.62 \times 10^{-45}$ | $8.62 \times 10^{-83}$ |

Overall, there were differences in the means between the groups; hence, the signals could be classified into segments, which reinforced the idea that environmental factors influence EEG signals. Interestingly, in the validation state, it was reported that the participants were most comfortable in events 1, 2, 8, and 9 and least comfortable in events 3 and 4. Therefore, it is important to take into account that channels Fp1, O1, and O2 might be significant because channels O1 and O2 in the occipital lobe are related to vision. This is due to the fact that the participants did mention that they felt less comfortable when the light was turned off (events 3, 4, 5, and 6 had lower self-reported scores than the rest because the light was off).

The grand mean topography of the EEG signal features extracted from the ANT Neuro and DIY EEG data is presented in Figure 5 along with the "difference" headmaps for comparison. Across different features, there are significant differences in channel P5 (in three features), O2 (in three features), and P6 (in three features), while the grand means for channels Fp2, C3, and C4 are similar between the two types of devices. For skewness and detrended fluctuation analysis, the distributions are somewhat different. However, for the remaining features, both the ANT Neuro and the DIY EEG data were markedly larger over the left-central and right-parietal sites and smaller at the right-frontal and right-central sites.

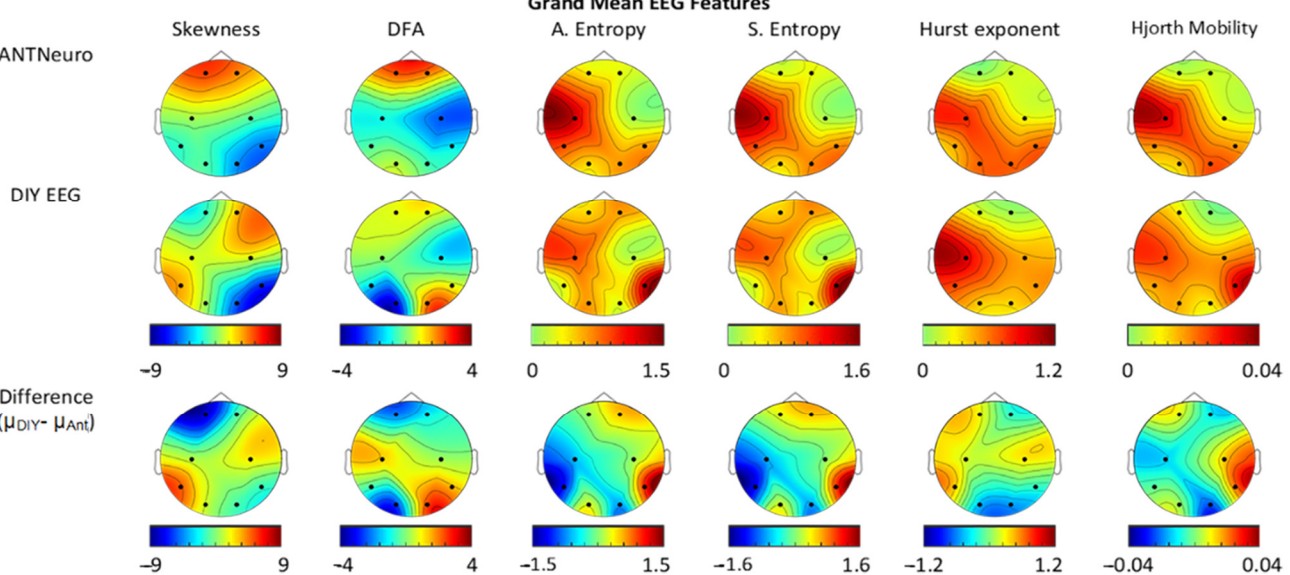

**Figure 5.** Grand mean EEG feature headmaps.

Based on the impact of how the skewness of each channel affects the prediction of the model, it can be seen from Figures 6 and 7 that the DIY headset has a degree of accuracy similar to that of the ANT Neuro headset (0.947 compared to 0.967).

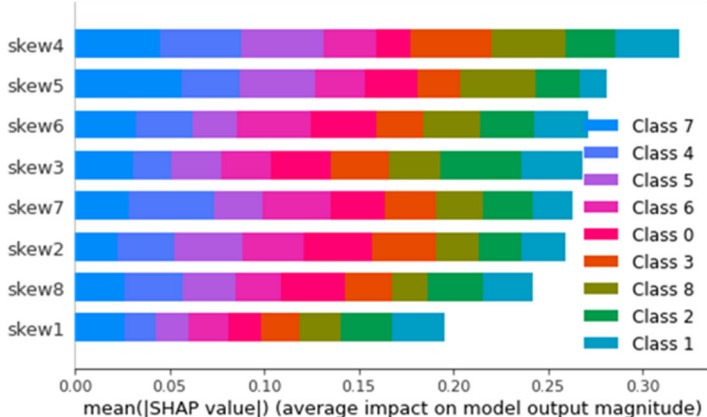

**Figure 6.** SHAP value impact on model prediction for ANT Neuro headset in terms of skewness.

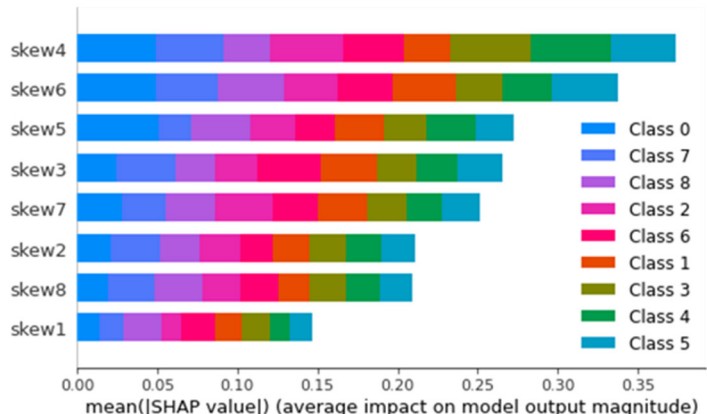

**Figure 7.** SHAP value impact on model prediction for DIY headset in terms of skewness.

Based on the impact of how the approximate entropy of each channel affects the prediction of the model, it can be seen from Figures 8 and 9 that the DIY headset has a degree of accuracy similar to that of the ANT Neuro headset (0.974 compared to 0.981).

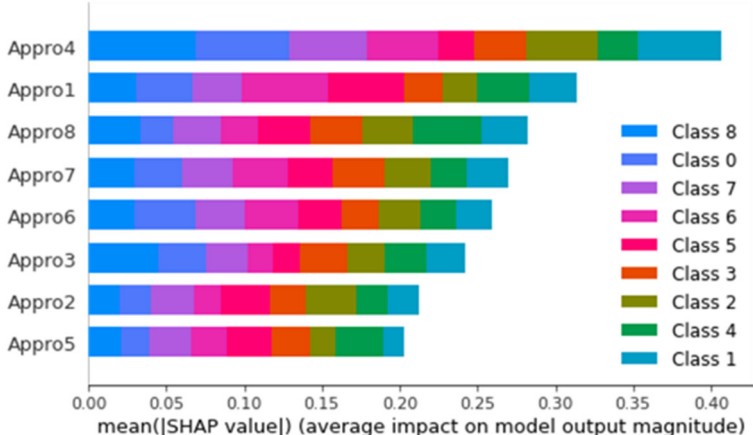

**Figure 8.** SHAP value impact on model prediction for ANT Neuro headset in terms of approximate entropy.

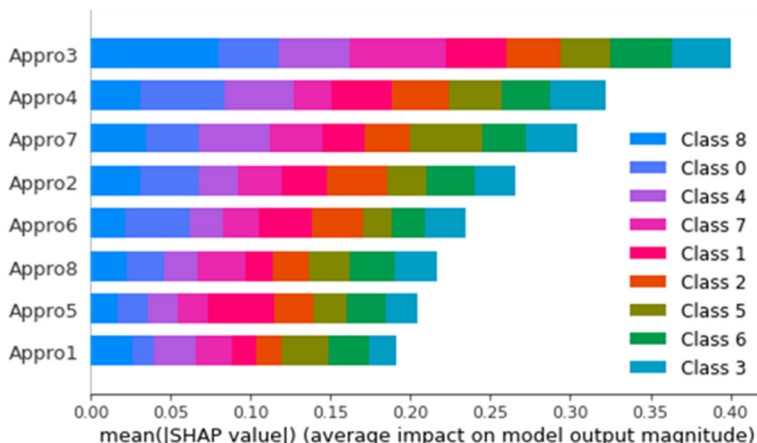

**Figure 9.** SHAP value impact on model prediction for DIY headset in terms of approximate entropy.

Based on the impact of how the sample entropy of each channel affects the prediction of the model, it can be seen from Figures 10 and 11 that the DIY headset has a degree of accuracy similar to that of the ANT Neuro headset (0.978 compared to 0.984).

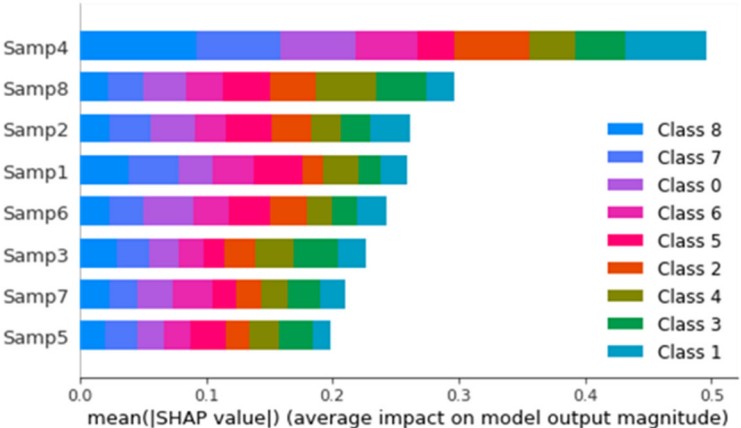

**Figure 10.** SHAP value impact on model prediction for ANT Neuro headset in terms of sample entropy.

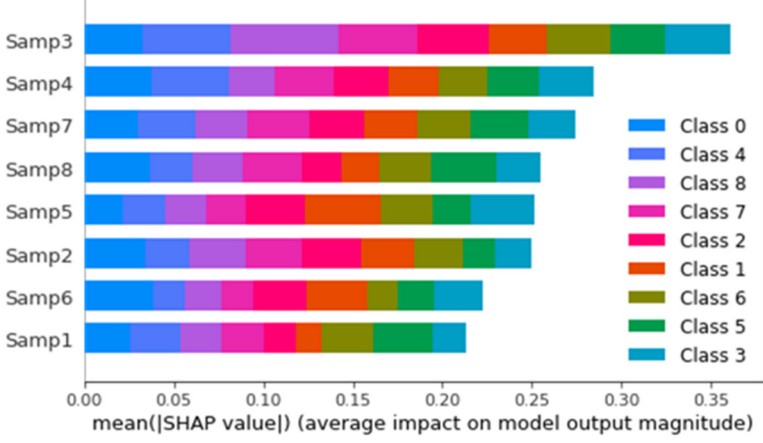

**Figure 11.** SHAP value impact on model prediction for DIY headset in terms of sample entropy.

While the accuracy of how each factor was used to predict EEG data is high and almost similar for both models, the magnitudes of the impact of each channel on the output of the

model also shared similar trends in terms of important channels. This suggested that the EEG data from our DIY headset were valid and could be used to train machine learning models in our subsequent steps.

### 3.5. Data Analysis

### 3.5.1. Relationship between the Environment and Physiological Well-Being

The data collected were processed using Python standard libraries such as Numpy, Sklearn, and Pandas. Outlier detection was performed on the data collected using the z-score method to remove outlier data. Firstly, a linear correlation was drawn between the environmental and biometric factors. Next, a random forest regression model was trained on the data to find the non-linear connections between the environmental factors and biometric factors. Finally, to interpret the contribution and significance of each environmental factor to a specific biometric factor, Shapley summary plots were graphed to find more complex relationships between the environmental factors and the biometric factors.

### 3.5.2. Pre-Processing and Feature Extraction of EEG Time Series

The EEG time series was preprocessed using bandpass and bandstop filters, the reference electrode standardization technique (REST), and the independent component analysis (for filtering out artifacts in the feature extraction model (Figure 12)). The EEG time series were cut into windows (bins) and used to predict the immediate output sample before (or after) it. In the feature extraction model, five bands of spectral information of 0.5 Hz, 4 Hz, 7 Hz, 12 Hz, 30 Hz, and 100 Hz were used.

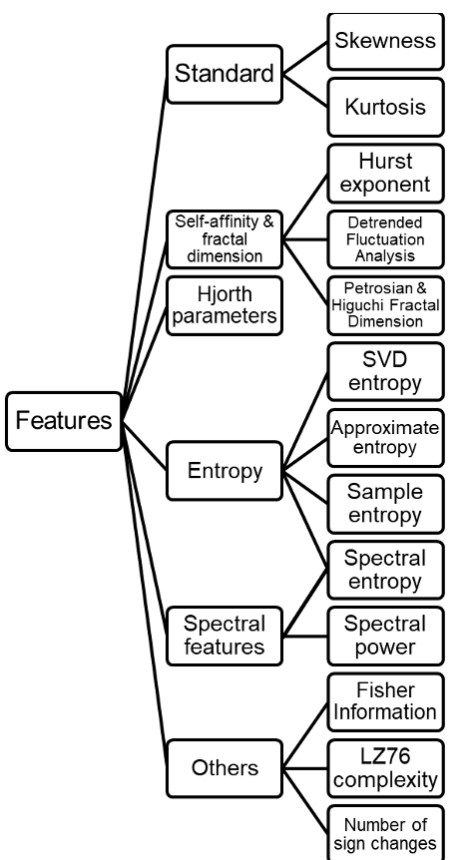

**Figure 12.** List of features extracted and used from collected EEG time series.

In this paper, different features were extracted from the EEG time series, which were used for the random forest regression model. The Hjorth parameters, activity and

complexity, which are indicators of the statistical properties used in signal processing in the EEG time domain, were used.

Hjorth activity represents the signal power and the variance of a time function and can indicate the surface of the power spectrum in the frequency domain.

Hjorth complexity represents the change in frequency. The parameter compares the similarity of the signal to a pure sine wave, where the value converges to 1 if the signal is more similar.

Spectral entropy or the energy spectral density is also used in the feature extraction model, which describes how the energy of a signal or a time series is distributed with the frequency.

### 3.5.3. Associations between the Functioning of Brain (EEG) and Physiological Well-Being and Associations between the Functioning of Brain (EEG) and the Environment

After the preprocessing of the data and the analysis, as described in the preceding sections, both the CNN and the feature extraction random forest regression models were used to analyze the associations between the functioning of brain (EEG) and physiological well-being, with the former model being generally higher in accuracy. For the analysis of the different associations between the functioning of brain (EEG) and the environment, only the CNN models were used. The results of the random forest regression models were interpreted using Shapley values and Shapley summary plots, while the CNN model results were interpreted using saliency maps, Shapley value maps, and frequency–amplitude gradient (FAG) maps.

### 4. Results

More than 100,000 data entries were collected from July 2021 to June 2022; after cleaning, 43,434 data points were used. Table 2 presents the descriptive statistics for the environmental variables; Table 3 presents those for the biometric variables; and Table 4 presents those for the data collected on the stress score. The reference values for the carbon dioxide concentration are approximately 400 ppm, for the PM2.5 dust it is the 10 $\mu g/m^3$ annual mean, and the atmospheric pressure at sea-level is 1013.25 millibars [18–20,31–33]. From the historical records, the diurnal temperature range of Singapore is 25 deg C to 33 deg C and the relative humidity in the island nation ranges from 60 to 90%, typically. From June to September, the climate of Singapore is influenced by the southwest monsoon, after which is the inter-monsoonal period of relatively weaker winds.

**Table 2.** Summary of collected environmental data.

| | Sound | Visible Light | Infrared Rad | Ultra-Violet | Temp | Rel Hum | Press-Ure | $CO_2$ Conc | Dust Conc | Wind Direction | Wind Speed |
|---|---|---|---|---|---|---|---|---|---|---|---|
| Min | 1.00 | 217.00 | 50.00 | 0.00 | 19.64 | 36.00 | 21.00 | 92.00 | 1.00 | N.A. | 2.80 |
| Max | 780.00 | 3327.00 | 9265.00 | 227.93 | 38.04 | 94.00 | 1474.00 | 4458.00 | 806.00 | N.A. | 16.60 |
| Mean | 8.72 | 261.56 | 259.58 | 0.06 | 29.67 | 70.15 | 617.11 | 517.06 | 25.45 | SW | 5.71 |
| Std dev | 20.57 | 33.26 | 83.51 | 2.09 | 1.55 | 5.60 | 235.35 | 170.82 | 52.11 | 123.12 | 1.26 |

The data were analyzed using the Python libraries Sklearn and SHAP. Correlations were drawn based on a best-fit line graph between each respective microclimatic and biometric/well-being variable.

Random forest regression models were then performed. Random forest regression models are supervised learning algorithms that use ensemble learning methods for regression. In turn, an ensemble learning method is a technique that combines predictions from multiple machine learning algorithms to make a more accurate prediction than a single model; in the case of the study reported, it used multiple regression models. Finally, outlier events were identified for subsequent investigation if necessary.

**Table 3.** Summary of collected biometric data.

| Features | Biometric Variables | Features | Biometric Variables |
|---|---|---|---|
| | Heart Rate | | Heart Rate |
| N | 43,434 | n | 43,434 |
| Min | 37.00 | Min | 37.00 |
| Max | 169.00 | Max | 169.00 |
| Mean | 81.17 | Mean | 81.17 |
| Std dev | 18.98 | Std dev | 18.98 |

**Table 4.** Summary of data collected on stress score.

| | Stress Score |
|---|---|
| N | 3440 |
| Min | 5.00 |
| Max | 90.00 |
| Mean | 23.48 |
| Std dev | 20.74 |

Table 5 shows the scaled data regression generated from the combined data of all five devices. In the table, the data cells in the column 'score' contain an R-squared score that is calculated from the regression data analysis from Python, while those in the other columns represent the respective feature importance of each input. R-squared is a statistical measure that represents the goodness of fit of a regression model. The ideal value for R-squared is 1. The closer the value of R-squared is to 1, the better the model fit. The higher this score is, the stronger the connection between the independent variable (namely the microclimate, in the case of the study reported) and the dependent variable (namely the biometric and well-being data).

**Table 5.** Goodness of fit from random forest regression analysis.

| Biometric Factors (Model Output) | Score | Environmental Variables (Model Input) | | | | | | | | | | |
|---|---|---|---|---|---|---|---|---|---|---|---|---|
| | | Sound | Visible Light | Infrared Rad | Ultra-violet | Temp | Rel Hum | Pres-sure | $CO_2$ Conc | Dust Conc | Wind | Time |
| Heart rate | 0.73 | 0.14 | 0.03 | 0.05 | 0.02 | 0.08 | 0.06 | 0.08 | 0.15 | 0.08 | 0.10 | 0.21 |
| rMSSD | 0.79 | 0.46 | 0.03 | 0.03 | 0.03 | 0.07 | 0.04 | 0.05 | 0.05 | 0.09 | 0.11 | 0.05 |
| Infrared | 0.51 | 0.08 | 0.05 | 0.06 | 0.06 | 0.09 | 0.08 | 0.10 | 0.07 | 0.10 | 0.13 | 0.18 |
| Skin temp | 0.72 | 0.21 | 0.04 | 0.04 | 0.02 | 0.12 | 0.06 | 0.08 | 0.11 | 0.13 | 0.10 | 0.08 |
| Stress | 0.85 | 0.14 | 0.05 | 0.05 | 0.01 | 0.05 | 0.04 | 0.03 | 0.49 | 0.04 | 0.05 | 0.04 |

To measure the respective contributions of the various predictors (the microclimate variables) against the actual values, Shapley summary plots were generated from the training data. Shapley values can be thought of as the average of the marginal contributions across all the possible permutations within a given model. Simply put, Shapley values decompose a prediction to show the impact of each feature by showing how much each feature contributed to the overall predictions.

The sections below elaborate on the respective contributions of each of the microclimatic variables measured in this study to each of the biometric indicators and indicators of well-being of interest. Within each section, two graphs are presented, the first being a graph generated by the random forest regression model pertaining to the given biometric/well-

being indicator and the second being a Shapley summary plot showing the contributions of the various microclimate variables to the given biometric/well-being indicator.

In the Shapley plots, the variables are ranked in descending order of importance, and the situation of each dot along the x-axis shows whether the effect of that value is associated with a higher or lower prediction. Simple exponential smoothing was used on the models. Each environmental feature had its smoothed counterpart, with the format: "(feature)_", which was derived from the values of that feature from the preceding hours, with older values being exponentially less important than the current value. For example: for f(t), we have f(t − 1), f(t − 2), ... affecting it, but f(t − 1) will be of more importance to f(t) than f(t − 2), f(t − 3), ... exponentially.

The features "0", "1", ... , "23" represent the hours of the day that the data points were recorded, while "31" to "37" are the days of the week (Monday to Sunday), which correspond to the time of recording of those data points. These features are binary; so, 1 or high means true, and 0 or low means false. For example, if a data point was recorded at 6 AM on a Monday, then "6" and "31" would be 1, and the remaining binary features would be 0. Finally, the color shows whether that variable is high (red) or low (blue) for that observation. In this way, Shapley summary plots combine feature importance with feature effects.

*4.1. Associative Relationships of Microclimate and Physical Health*

4.1.1. Heart Rate

Figure 13 shows the graph of the predicted values against the true values of the variable 'heart rate', and Figure 14 shows the Shapley summary plot of the associations of the microclimate with the same variable.

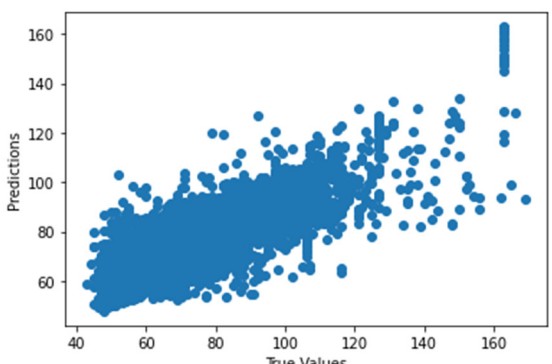

**Figure 13.** Predicted values against true values of heart rate.

From Figure 14, the variable 'heart rate' is most strongly associated with the variable 'carbon dioxide concentration' (feature importance of 0.15), followed by the variable 'sound' (feature importance of 0.14), and by the three variables which affect heart rate co-equally (feature importance of 0.08), namely 'ambient temperature', 'dust concentration', and 'air pressure'. From the above Shapley summary plot, the lower the carbon dioxide concentration, the lower the heart rate. As for sound, the higher the sound in the surrounding microclimate is, the higher the heart rate level. Similarly, the higher the pressure and dust concentration, the higher the heart rate level, while the heart rate value decreases when the ambient temperature increases. An interesting observation is that the heart rate dropped from 5 pm onward.

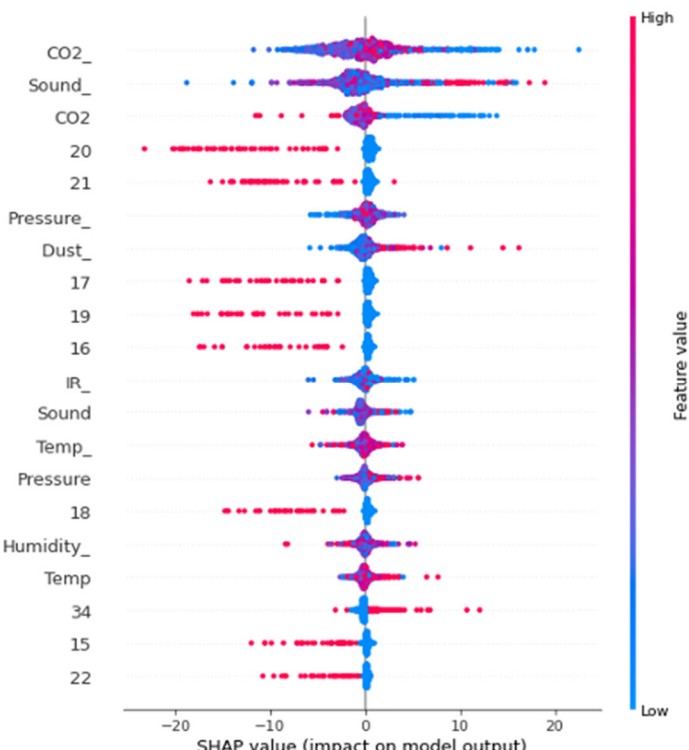

**Figure 14.** Shapley summary plot of microclimatic factors and heart rate.

4.1.2. Heart Rate Variability (rMSSD)

In the context of heart rate variability, rMSSD refers to the physiological phenomenon of variation in the time interval between heartbeats. Figure 15 shows the graph of predicted values against the true values of the rMSSD, and Figure 16 shows the Shapley summary plot of the associations of the microclimate with the same variable.

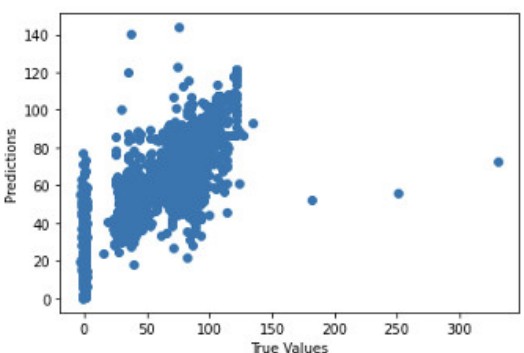

**Figure 15.** Predicted values against true values of rMSSD.

From Figure 16, the variable 'root mean square of successive beat-to-beat interval differences' ('rMSSD') is most strongly associated with the variable 'sound' (feature importance of 0.46), followed by the variable 'dust concentration' (feature importance of 0.09), then co-equally (feature importance of 0.07) by the variables 'ambient temperature' and 'wind direction'. From the above Shapley summary plot, the higher the sound is, the lower the rMSSD. The higher dust concentration, the lower the rMSSD. Similarly, the higher the ambient temperature and the more westerly the wind, the lower the rMSSD.

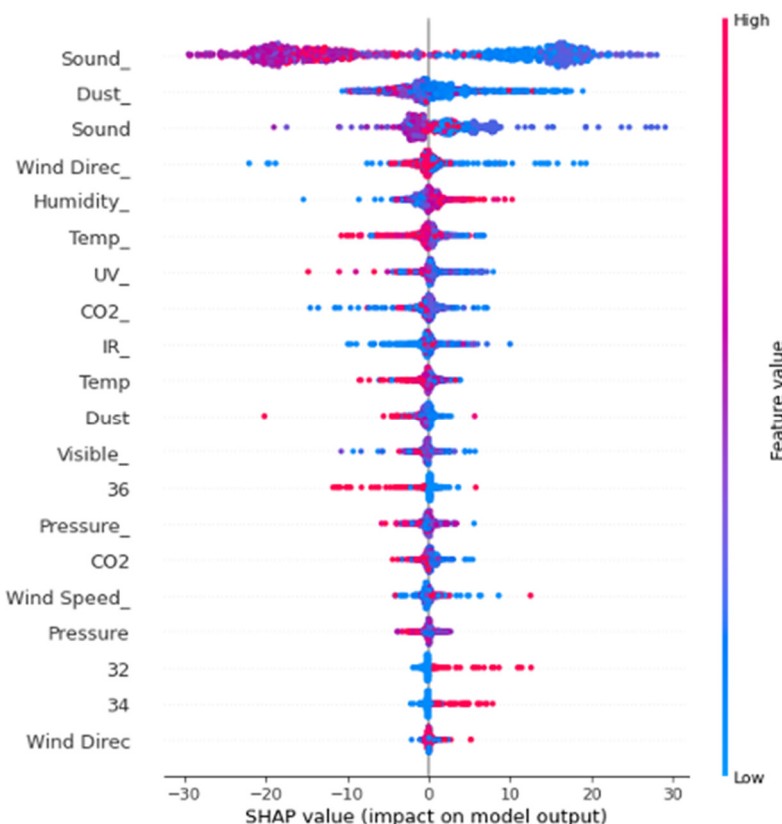

**Figure 16.** Shapley summary plot of microclimatic factors and rMSSD.

### 4.1.3. Skin Temperature at the Wrist

Figure 17 shows the graph of the predicted values against the true values of the variable 'skin temperature', and Figure 18 shows the Shapley summary plot of the associations of the microclimate with the same variable.

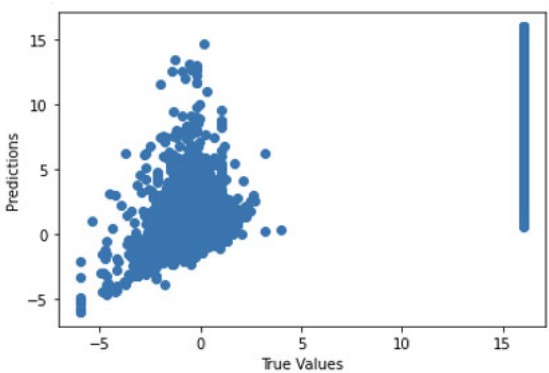

**Figure 17.** Predicted values against true values of skin temperature.

From Figure 18, the variable 'skin temperature at the wrist' is most strongly associated with the variable 'sound' (feature importance of 0.21), followed by the variable 'dust concentration' (feature importance of 0.13) and the variable 'ambient temperature' (feature importance of 0.12). From the above Shapley summary plot, the higher sound, the higher the skin temperature. In attempting to interpret this, one might consider sound to be a proxy indicator of the nature of one's immediate environment, which—in turn—may have attributes which result in physiological responses, including with respect to skin temperature. The lower the amount of dust concentration that is present, the higher the

skin temperature. The higher ambient temperature, the higher the skin temperature, but it rises at a lower rate.

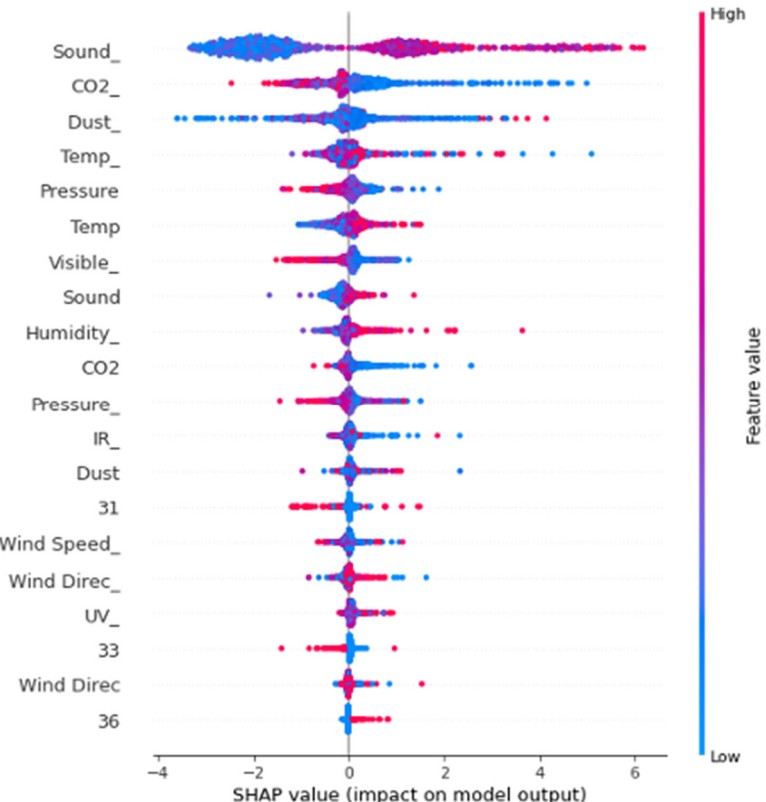

**Figure 18.** Shapley summary plot of microclimatic factors and skin temperature.

### 4.2. Associative Relationships of Microclimate and Stress Score

The variable 'stress score' is generated by the smartwatches and is derived from oxygen saturation and heart rate. Figure 19 shows the graph of the predicted values against the true values of the variable 'stress score', and Figure 20 shows the Shapley summary plot of the associations of the microclimate with the same variable.

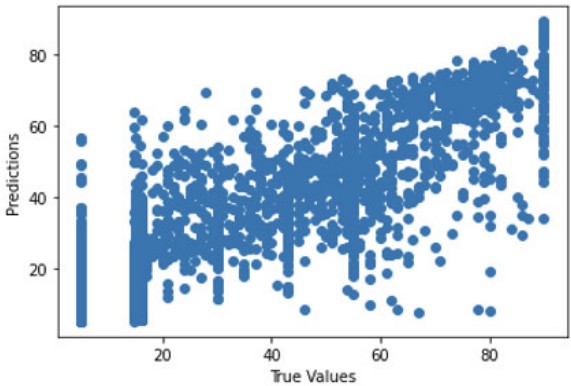

**Figure 19.** Predicted values against true values of stress score.

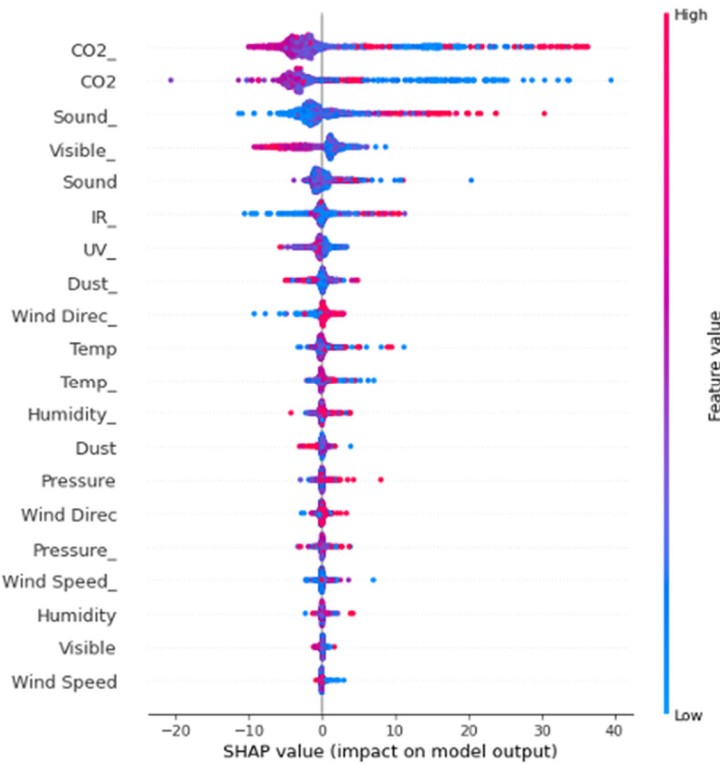

**Figure 20.** Shapley summary plot of microclimatic factors and stress score.

From Figure 20, the variable 'stress score' is most strongly associated with the variable 'infrared radiation' (feature importance of 0.363), followed by the variable 'carbon dioxide concentration' (feature importance of 0.226) and by the variable 'ambient temperature' (feature importance of 0.104). The higher the infrared radiation, the higher the stress level. As for carbon dioxide concentration, the higher the concentration, the higher the stress level. For ambient temperature, the lower the temperature, the lower the stress level.

*4.3. Associative Relationships of EEG Data (Brainwaves) and Physical Health*

4.3.1. Heart Rate

Feature Extraction Random Forest Regression Model

Figure 21 shows the graph of the predicted values against the true values of the variable 'heart rate', and Figure 22 shows the Shapley summary plot of the associations of the EEG with the same variable.

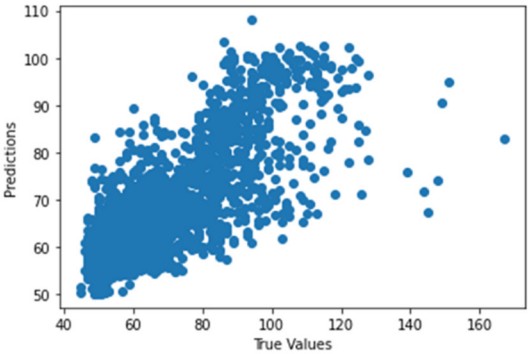

**Figure 21.** Predicted values against true values of feature extraction random forest regression model of EEG—heart rate ($R^2$ score = 0.58).

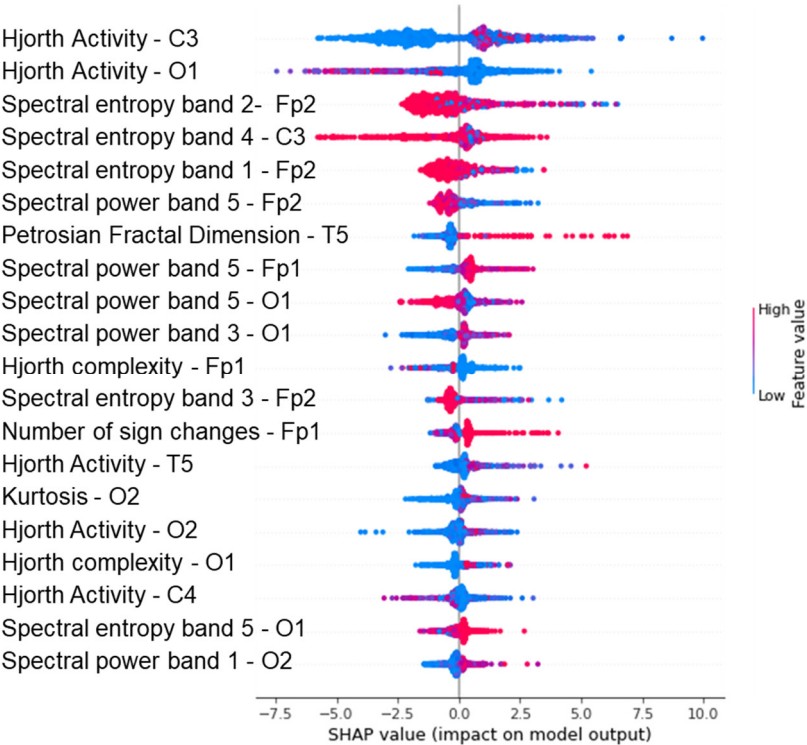

**Figure 22.** Shapley summary plot of feature extraction random forest regression model of EEG—heart rate.

From Figure 22, the variable 'heart rate' is most strongly associated with the variable 'Hjorth Activity' (channel C3, O1), followed by the variable 'Spectral entropy band' (band 3—alpha waves, 5—delta waves, and 1—gamma waves). From the above Shapley summary plot, the high Hjorth activity of channel C3 leads to an increase in heart rate, and the low Hjorth activity of channel O1 leads to a decrease in heart rate. In terms of spectral power, for spectral power band 5, a higher spectral power around region Fp2 and O1 is strongly associated with a decrease in heart rate, while a higher value at region Fp1 is strongly associated with an increase in heart rate. For spectral power band 3, a higher spectral power around region O1 is strongly associated with an increase in heart rate. For spectral power band 1, a higher spectral power around region O1 is strongly associated with an increase in heart rate. Therefore, it can be seen that the Alpha, Delta, and Gamma waves are strongly associated with changes made to the heart rate. Furthermore, it is observed that the most influential channels are Fp2, O1, O2, and Fp1. As high cognitive functions are associated with higher heart rate variability [34] and heart rate variability is inversely correlated to heart rate [35], the frontal and occipital lobes are most related to heart rate, which generally suggests that high cognitive functions lead to a lower heart rate.

Convolutional Neural Network Model

Figure 23 shows the graph of the predicted values against the true values of the variable 'heart rate', and Figure 24 shows the results of the CNN model with the same variable.

From Figure 24, the most relevant and significant channels are Fp2, O1, and Fp1, which suggest the high cognitive functions. In the saliency map, channels C3 and C4 are also significant in leading to the CNN predictions. This suggests that the temporal lobes, which are responsible for processing affect/emotions, language, and certain aspects of visual perception, are a major weight on brain activity. In the amplitude gradient, there is a high gradient at around 20 Hz and 83 Hz, which are in the beta and gamma bands. This indicates conscious focus, improved memory, and problem solving.

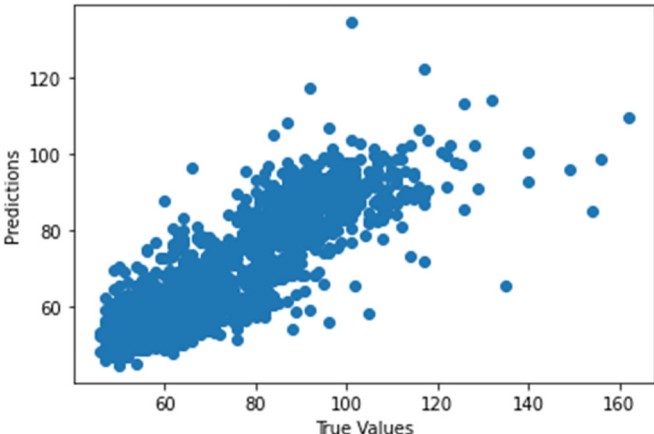

**Figure 23.** Predicted values against true values of CNN model of EEG—heart rate ($R^2$ score = 0.72).

**Figure 24.** Figures for EEG–heart rate CNN model results.

### 4.3.2. Body Temperature

Feature Extraction Random Forest Regression Model

Figure 25 shows the graph of the predicted values against the true values of the variable 'body temperature', and Figure 26 shows the Shapley summary plot of the associations of EEG with the same variable.

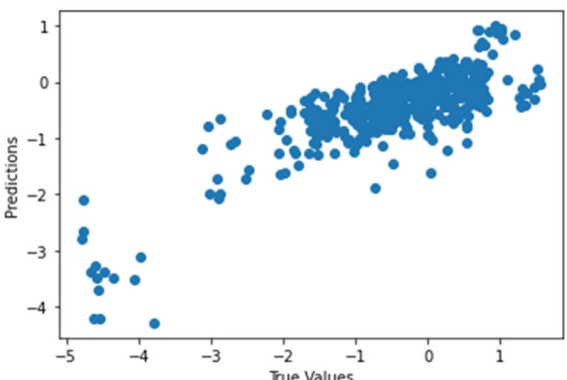

**Figure 25.** Predicted values against true values of feature extraction random forest regression model of EEG—body temperature ($R^2$ score = 0.63).

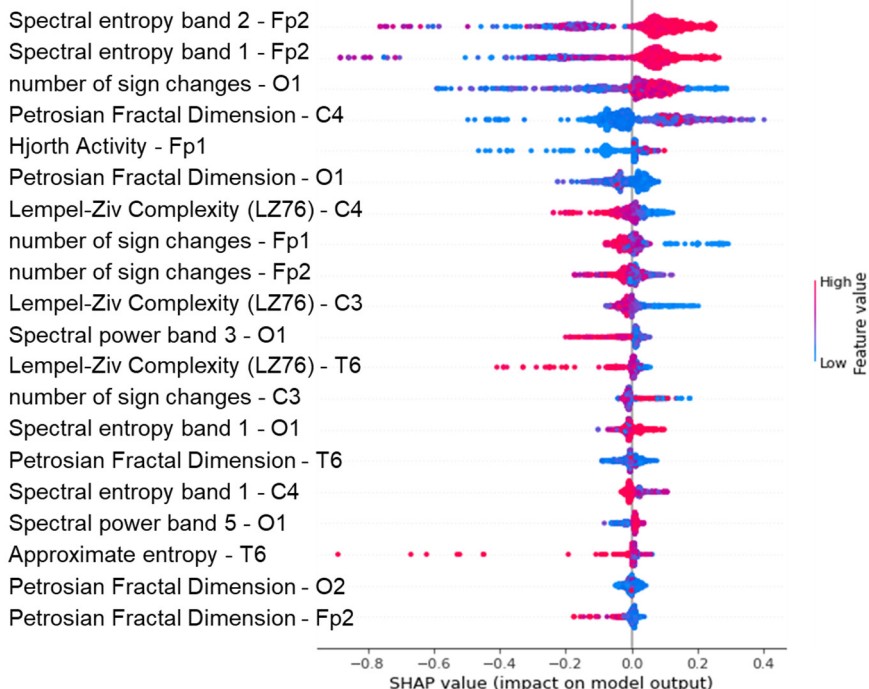

**Figure 26.** Shapley summary plot of feature extraction random forest regression model of EEG—body temperature.

From Figure 26, the variable 'body temperature' is most strongly associated with the variable 'Spectral entropy' (bands 1 and 2), followed by the variable 'Number of sign changes' (channels O1, Fp1, Fp2 and C3). From the above Shapley summary plot, in terms of spectral entropy, for spectral entropy band 1, a higher spectral entropy around region Fp2 and O1 is strongly associated with an increase in body temperature, while a higher value at region C4 is strongly associated with a decrease in body temperature. For spectral entropy band 2, a higher spectral power around region Fp2 is strongly associated with an increase in body temperature. Next, in terms of 'Numbers of sign changes', higher numbers of sign changes in channels Fp1 and Fp2 are associated with a decrease in body

temperature, while higher numbers of sign changes in channels O1 and C3 are associated with an increase in body temperature. For body temperature, the most significant channels are Fp2, O1, C4, Fp1, and C3, which suggests that high cognitive functions in the frontal and occipital lobes are strongly associated with increased body temperature. This is further supported by Kazama et al., 2012 [36].

Convolutional Neural Network Model

Figure 27 shows the graph of the predicted values against the true values of the variable 'body temperature', and Figure 28 shows the results of the CNN model with the same variable.

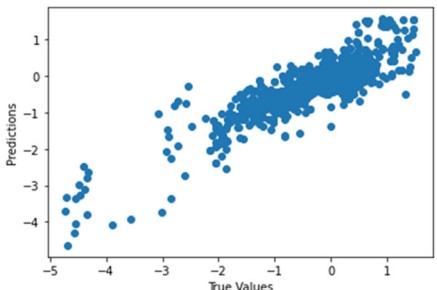

**Figure 27.** Predicted values against true values of CNN model of EEG—body temperature ($R^2$ score = 0.75).

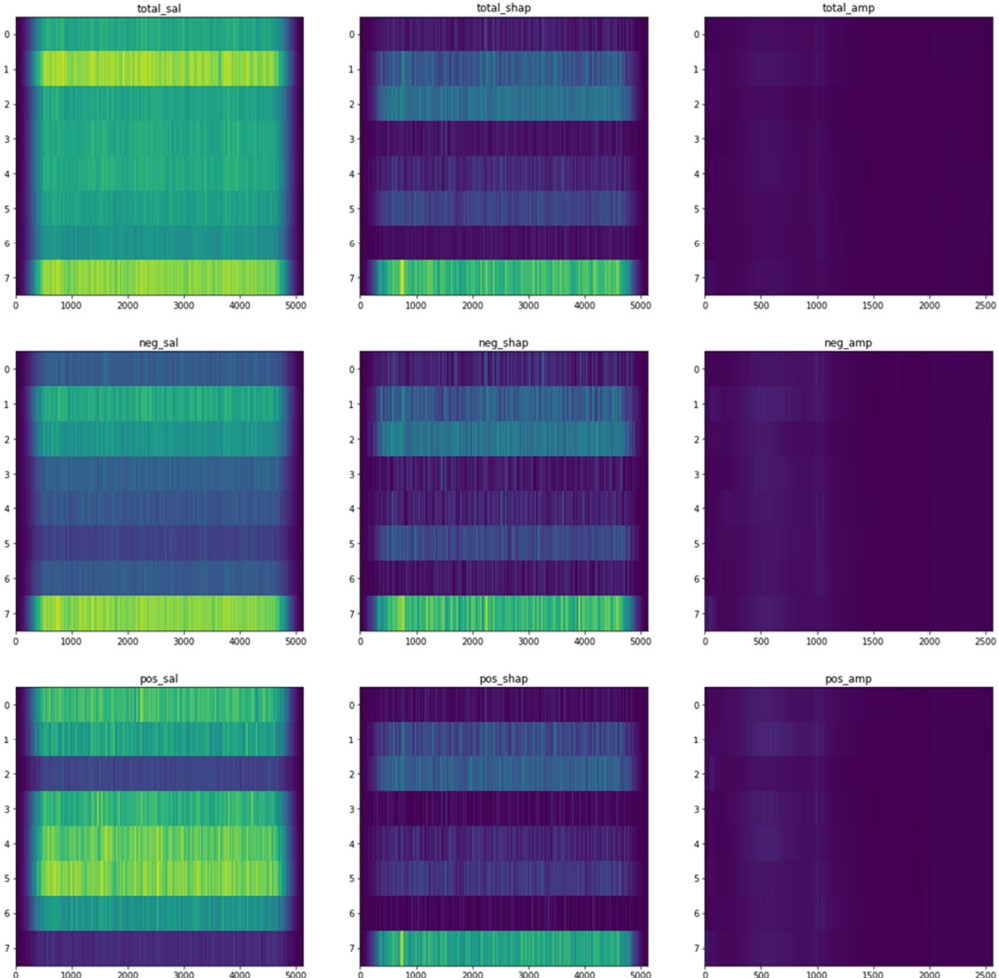

**Figure 28.** Figures for EEG—body temperature CNN model results.

From Figure 28, the most relevant and significant channels are Fp2, O2, C3, and T6, which suggest high cognitive functions. The fluctuation in channel C3 positively affects the prediction, while the fluctuation in channel T5 negatively affects the prediction. In the amplitude gradient, there is a high gradient at around 4.6 Hz, 24 Hz, and 49 Hz, which are in the theta, beta, and gamma bands. This indicates that conscious focus, improved memory, and problem solving are again strongly associated with body temperature.

### 4.4. Associative Relationships of EEG Data (Brainwaves) and the Environment

4.4.1. Dust Concentration

Feature Extraction Random Forest Regression Model

Figure 29 shows the graph of the predicted values against the true values of the feature extraction random forest regression model of EEG—dust concentration, and Figure 30 shows the corresponding Shapley summary plot.

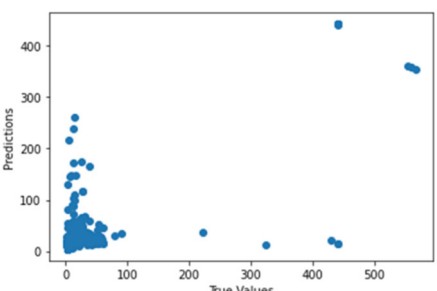

**Figure 29.** Predicted values against true values of feature extraction random forest regression model of EEG—dust concentration ($R^2$ score = 0.549).

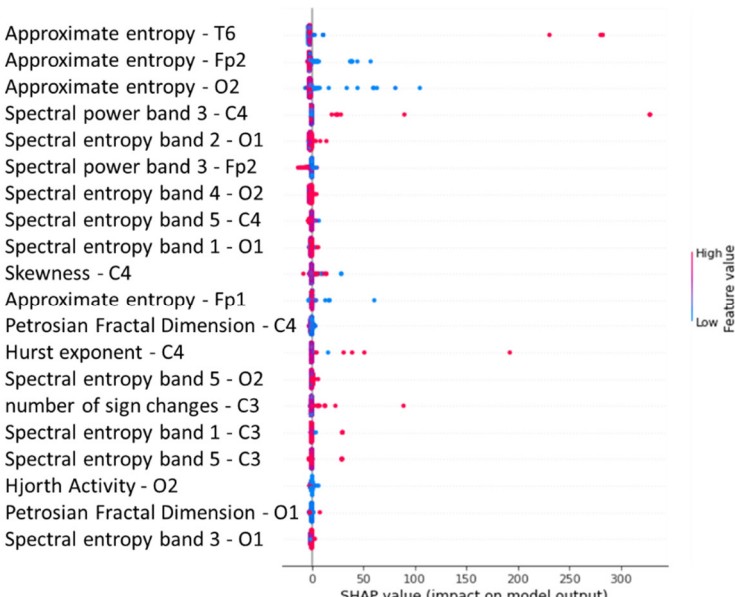

**Figure 30.** Shapley summary plot of feature extraction random forest regression model of EEG—dust concentration.

From Figure 30, the variable 'dust concentration' is most strongly associated with the variable 'Approximate entropy' (channels T6, Fp2 and O2), followed by the variable 'Spectral power' (band 3). From the above Shapley summary plot, in terms of approximate entropy, a lower value of the approximate entropy of the channels T6, Fp2, and O2 tends to decrease the dust concentration slightly, while a higher value of the approximate entropy of these channels has a stronger association with a more significant increase in dust concentration. In terms of the spectral power of band 3, a higher spectral power at channel

C4 is associated with an increase in dust concentration while a higher value of the spectral power at channel Fp2 is strongly associated with a decrease in dust concentration.

Convolutional Neural Network Model

Figure 31 shows the graph of the predicted values against the true values of the CNN model of the EEG—dust concentration, and Figure 32 shows the corresponding Shapley summary plot.

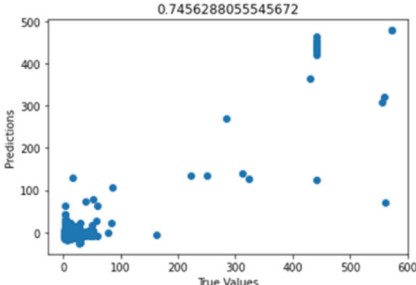

**Figure 31.** Predicted values against true values of CNN model of EEG—dust concentration ($R^2$ score = 0.746).

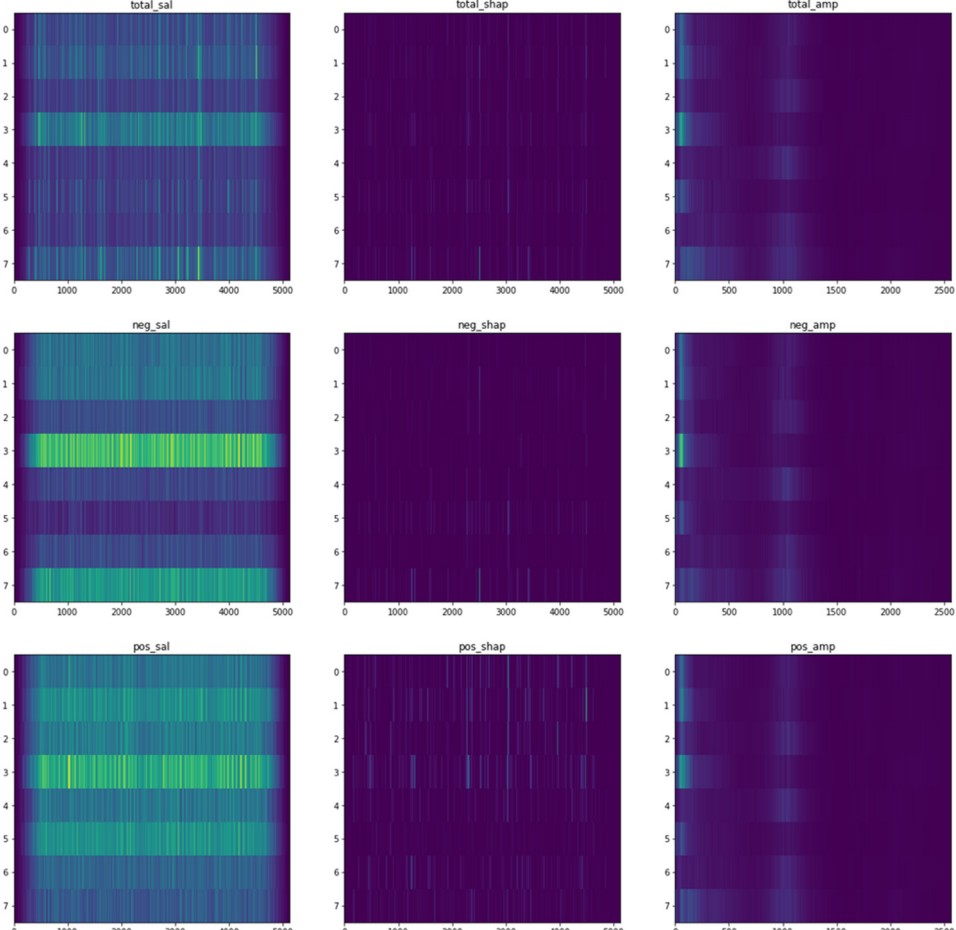

**Figure 32.** Figures for EEG—dust concentration CNN model results.

From Figure 32, the most relevant and significant channels are Fp2, C4, T6, and O2, which suggest high cognitive functions in the right brain hemisphere. From the negative saliency (neg_sal) time-series plot and the positive saliency (pos_sal) plot, the fluctuation in channel C4 generally negatively affects the prediction, while the fluctuation in channel

T6 positively affects the prediction. In the amplitude gradient, there is a high gradient at around 3.1 Hz, 4.7 Hz, and 50 Hz, which are in the delta, theta, and gamma bands. This indicates that a relaxed state of mind and deep sleep and, conversely, heightened perceptions and higher cognitive functions and processing are strongly associated with the dust concentration in the surrounding environment.

### 4.4.2. Ambient Temperature

Feature Extraction Random Forest Regression Model

Figure 33 shows the graph of the predicted values against the true values of the feature extraction random forest regression model of EEG—ambient temperature, and Figure 34 shows the corresponding Shapley summary plot.

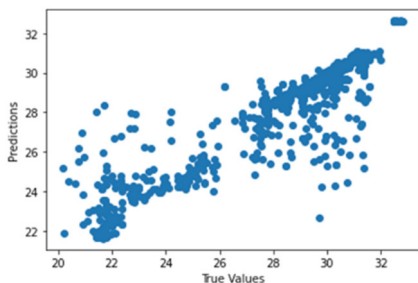

**Figure 33.** Predicted values against true values of feature extraction random forest regression model of EEG—ambient temperature ($R^2$ score = 0.791).

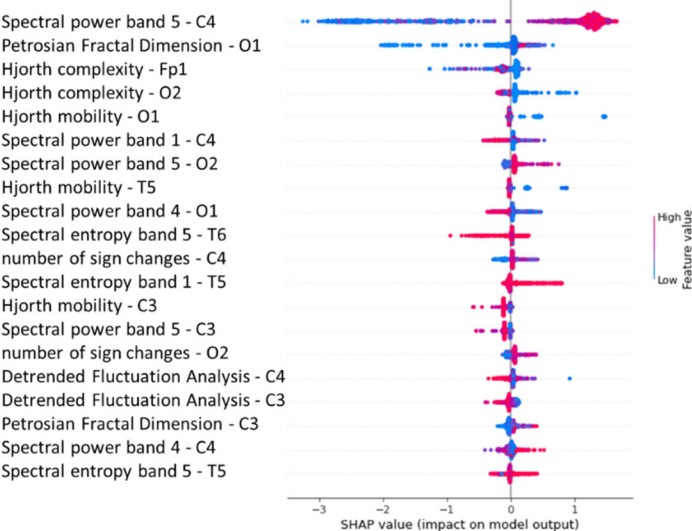

**Figure 34.** Shapley summary plot of feature extraction random forest regression model of EEG—ambient temperature.

From Figure 34, the variable 'ambient temperature' is most strongly associated with the variable 'Spectral power band 5' (channel C4), followed by the variable 'Petrosian Fractal Dimension' (channel O1) and Hjorth complexity (channel Fp1 and O2). From the above Shapley summary plot, in terms of the spectral power of band 5, a high value of the spectral power of channel C is strongly associated with an increase in ambient temperature and vice versa. In terms of the Petrosian fractal dimension of channel O1, a lower value of the Petrosian Fractal Dimension at channel O1 is associated with both a decrease and increase in ambient temperature. Finally, in terms of Hjorth complexity, a lower value of the Hjorth complexity of channel Fp1 tends to be associated with a decrease in ambient temperature, while a lower value of the Hjorth complexity of channel O2 tends to be associated with an increase in ambient temperature.

Convolutional Neural Network Model

Figure 35 shows the graph of the predicted values against the true values of the CNN model of EEG—ambient temperature, and Figure 36 shows the corresponding Shapley summary plot.

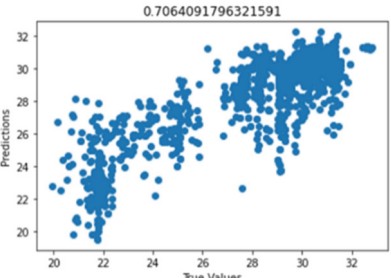

**Figure 35.** Predicted values against true values of CNN model of EEG—ambient temperature ($R^2$ score = 0.706).

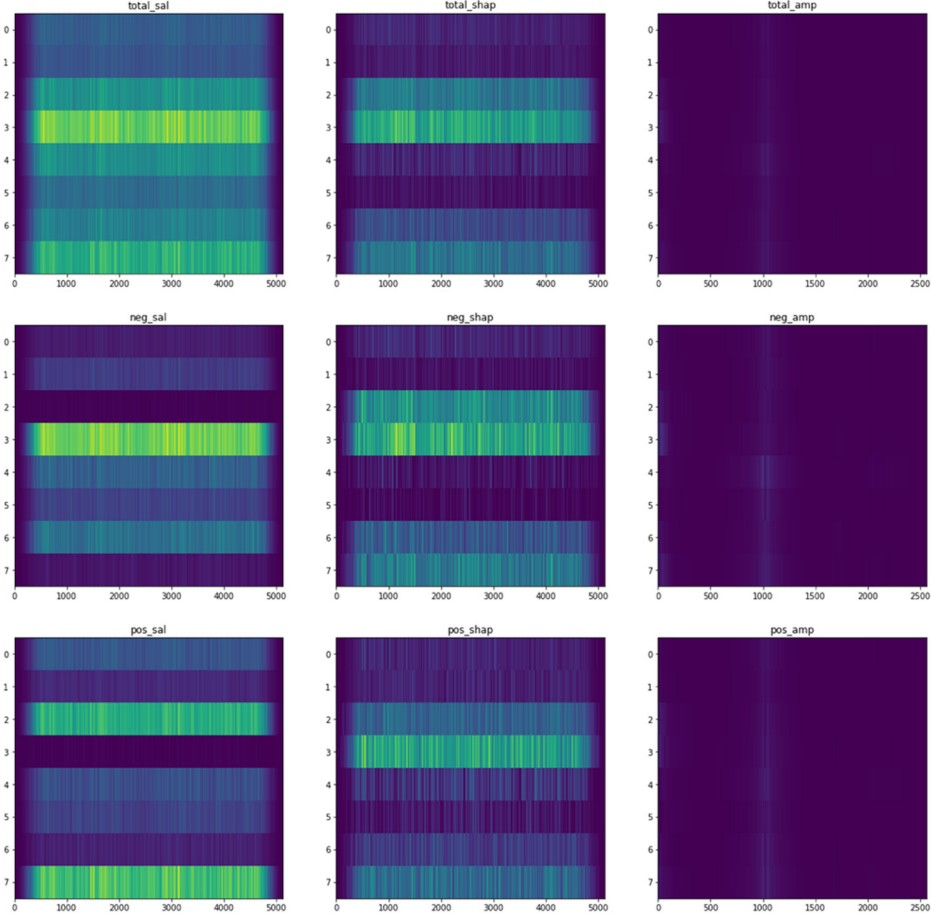

**Figure 36.** Figures for EEG—ambient temperature CNN model results.

From Figure 36, the most relevant and significant channels are C3, C4, T5, and O2, which suggest high cognitive functions, such as the visual, information processing, and voluntary movement. From the negative saliency (neg_sal) time-series plot and the positive saliency (pos_sal) plot, the fluctuation in channel C4 negatively affects the prediction, while the fluctuation in channel O2 positively affects the prediction. In the amplitude gradient, there is a high gradient at around 0.6 Hz and 52.5 Hz, which are in the delta and gamma bands. This indicates that deep sleep and heightened perceptions and higher

cognitive functions and processing are strongly associated with ambient temperature in the surrounding environment.

### 4.4.3. Carbon Dioxide Concentration

Feature Extraction Random Forest Regression Model

Figure 37 shows the graph of the predicted values against the true values of the feature extraction random forest regression model of EEG—$CO_2$ concentration, and Figure 38 shows the corresponding Shapley summary plot.

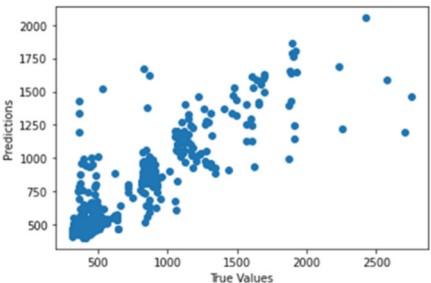

**Figure 37.** Predicted values against true values of feature extraction random forest regression model of EEG—$CO_2$ concentration ($R^2$ score = 0.763).

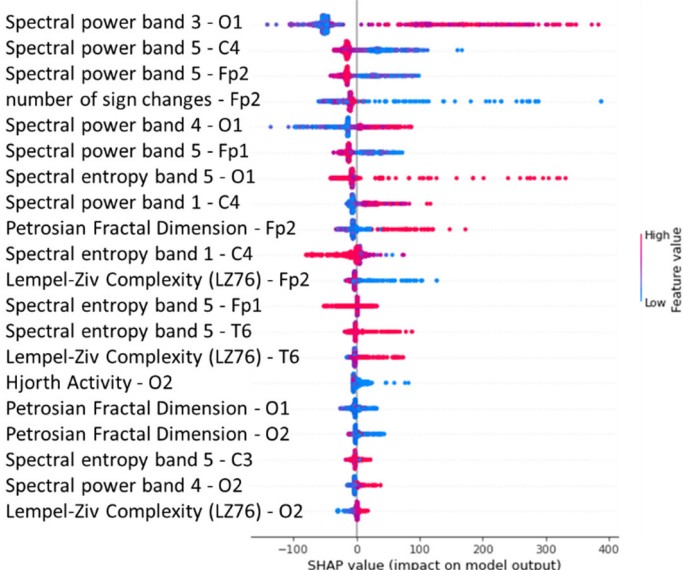

**Figure 38.** Shapley summary plot of feature extraction random forest regression model of EEG—$CO_2$ concentration.

From Figure 38, the variable 'carbon dioxide concentration' is most strongly associated with the variable 'Spectral power band' (bands 3 and 5), followed by the variable 'Number of sign changes' (channel Fp2). From the above Shapley summary plot, in terms of the spectral power band, a higher spectral power of band 3 is strongly associated with an increase in $CO_2$ concentration. For spectral entropy band 5, a higher spectral power around regions C4 and Fp2 is strongly associated with a decrease in $CO_2$ concentration. Next, in terms of 'Numbers of sign changes', lower number of sign changes in channel Fp2 are associated with an increase in $CO_2$ concentration.

Convolutional Neural Network Model

Figure 39 shows the graph of the predicted values against the true values of the CNN model of the EEG—$CO_2$ concentration, and Figure 40 shows the corresponding Shapley summary plot.

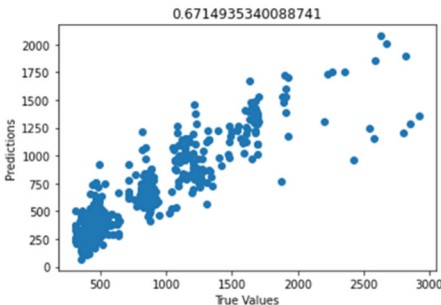

**Figure 39.** Predicted values against true values of CNN model of EEG—$CO_2$ concentration ($R^2$ score = 0.671).

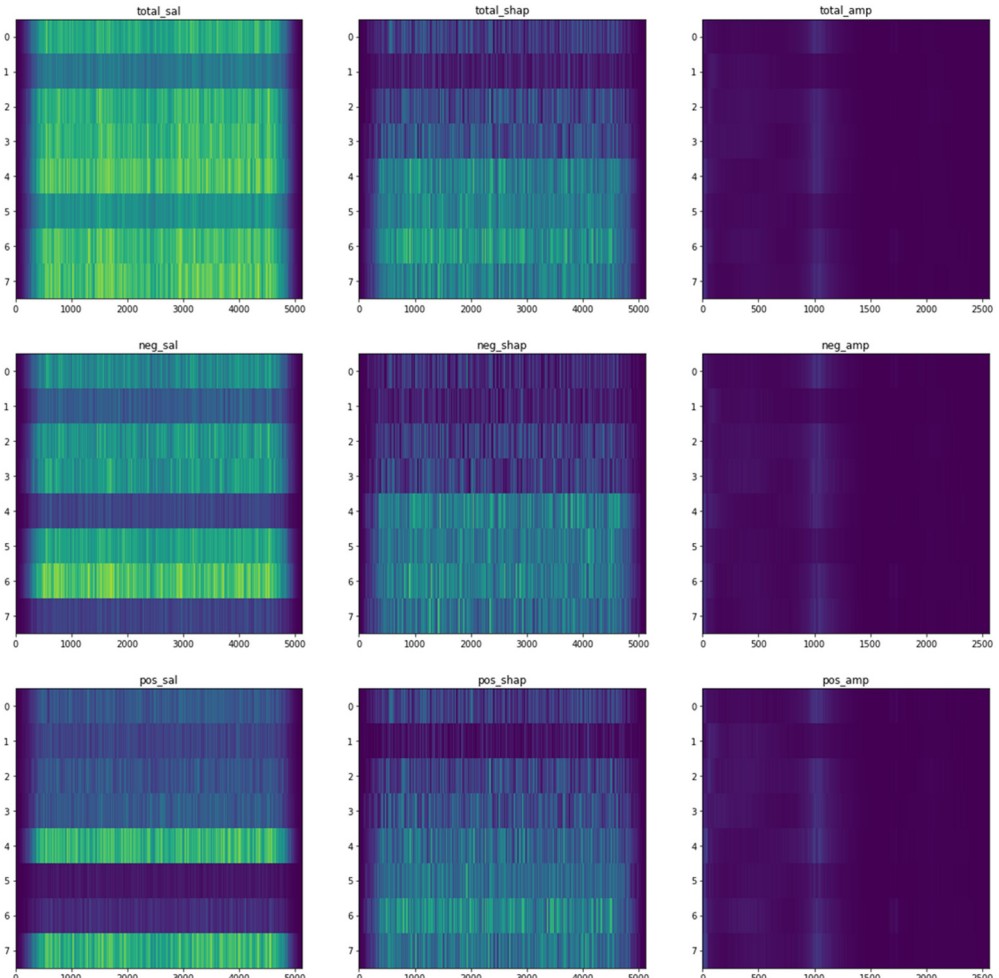

**Figure 40.** Figures for EEG—CO2 concentration CNN model results.

From Figure 40, the most relevant and significant channels are C4, T5, O1, and O2, which suggest high cognitive functions such as the visual, information processing and voluntary movement, according to the tasks participants are carrying out. From the negative saliency (neg_sal) time-series plot and the positive saliency (pos_sal) plot, the fluctuation in channel C4 negatively affects the prediction, while the fluctuation in channels T5 and O2 positively affects the prediction. In the amplitude gradient, there is a high gradient at around 0.6 Hz and 50 Hz, which are in the delta and gamma bands. This indicates that deep sleep and heightened perceptions and higher cognitive functions and processing are strongly associated with carbon dioxide concentration in the surrounding environment.

#### 4.4.4. Sound Intensity

Feature Extraction Random Forest Regression Model

Figure 41 shows the graph of the predicted values against the true values of the feature extraction random forest regression model of EEG—sound intensity, and Figure 42 shows the corresponding Shapley summary plot.

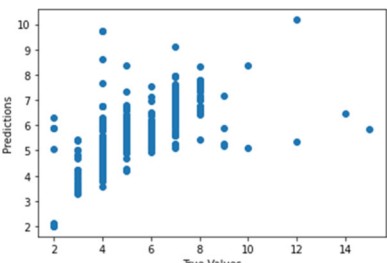

**Figure 41.** Predicted values against true values of feature extraction random forest regression model of EEG—sound intensity ($R^2$ score = 0.608).

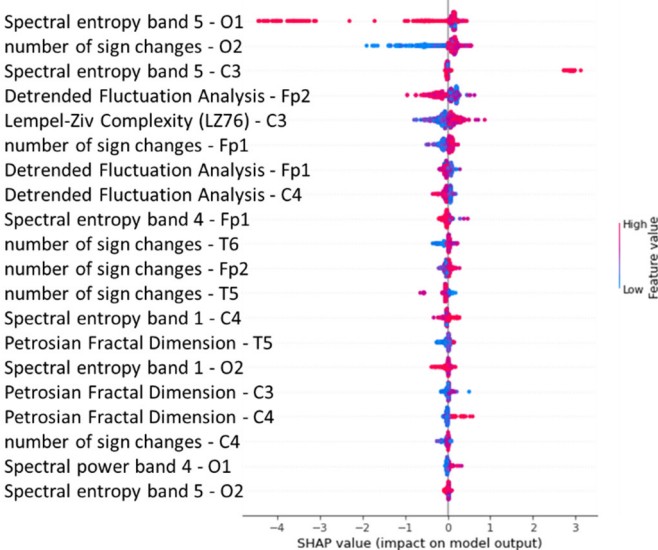

**Figure 42.** Shapley summary plot of feature extraction random forest regression model of EEG—sound intensity.

From Figure 42, the variable 'sound intensity' is most strongly associated with the variable 'Spectral entropy band 5' (channels O1 and C3), followed by the variable 'Number of sign changes' (channel O2). From the above Shapley summary plot, in terms of the spectral entropy of band 5, a higher spectral entropy of channel O1 is associated with a strong decrease in sound intensity, while a higher value is associated with a slight increase in sound intensity. For the spectral entropy of band 5 at region C3, a higher value of it is associated with an increase in sound intensity. Next, in terms of "number of sign changes", a lower value of sign changes at channel O2 is associated with a decrease in sound intensity.

Convolutional Neural Network Model

Figure 43 shows the graph of the predicted values against the true values of the CNN model of EEG—sound intensity, and Figure 44 shows the corresponding Shapley summary plot.

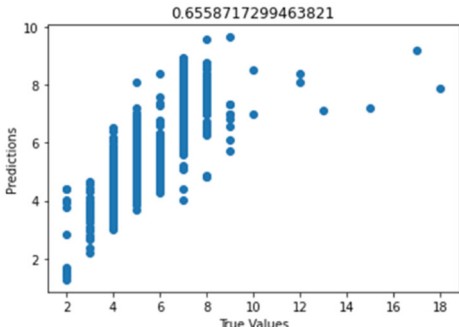

**Figure 43.** Predicted values against true values of CNN model of EEG—sound intensity ($R^2$ score = 0.656).

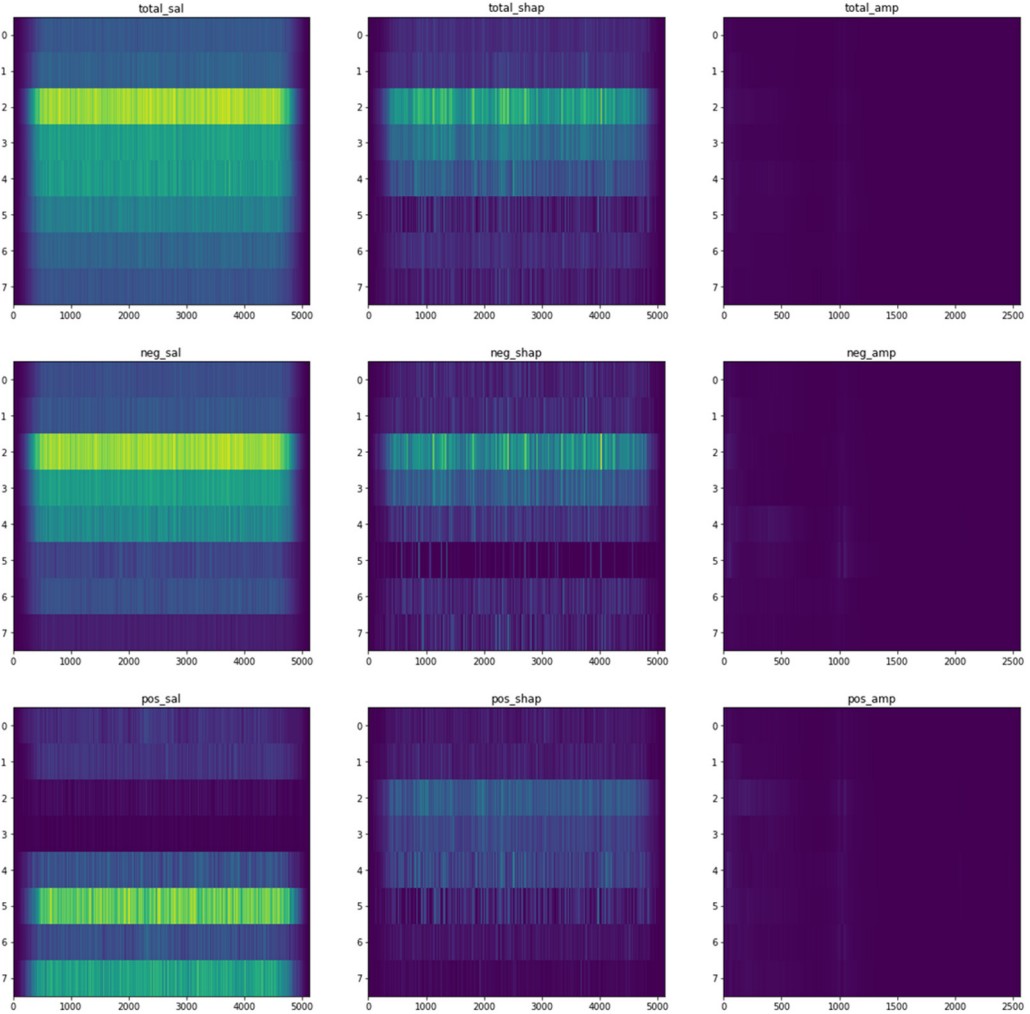

**Figure 44.** Figures for EEG—sound intensity CNN model results.

From Figure 44, the most relevant and significant channels are C3, C4, T5, and T6, which suggest high cognitive functions in terms of auditory stimuli, memory, and emotion. From the negative saliency (neg_sal) time-series plot and the positive saliency (pos_sal) plot, the fluctuation in channels C3 and C4 negatively affects the prediction, while the fluctuation in channels T6 and O2 positively affects the prediction. In the amplitude gradient, there is a high gradient at around 3.1 Hz,12.7 Hz, and 35.9 Hz, which are in the delta, alpha, and gamma bands. This indicates that deep sleep and the state of being mentally relaxed but

drowsy and heightened perceptions and higher cognitive functions and processing are strongly associated with sound intensity in the surrounding environment.

## 5. Discussion

Following a citizen research approach and manually designing and building our own EEG headset, the validation of the EEG data collected using the research grade equipment has proven to be fruitful as it reinforces the hypothesis that the environment is strongly associated with brain activities in a multi-dimensional and non-linear manner. In this research, we acknowledge that the extent of the effects of the environment are contingent solely upon the external factor, not taking into account the personal mood or feeling of the participants at the point of the conducting of the research. These might lead to assumptions and some degree of inaccuracy; however, the theory of environmental effects on the brain activities was suggested in past research. For example, Norwood et al. [37], in their work, have come to the conclusion that while natural environments were associated with low frequency brainwaves and lower brain activity in frontal areas, indicating comfortable and subjectively restorative feeling, urban environments, with a set of completely different environments and biomes, appear to induce brain responses associated with a negative effect. Certainly, the different ways in which humans can leverage the external settings of the environment to treat mental illnesses, neurological disorders, and to increase their concentration and productivity has already been studied.

The preceding analysis suggests that the more associative microclimatic factors are dust concentration, ambient temperature, carbon dioxide concentration, and sound intensity. These factors were continually associated with high feature importance scores in the EEG data signal and in both the objective scores recorded from the electronic instruments and the more subjective self-report forms. Furthermore, we have found that visual stimulus and problem processing, in terms of information, touch, and spatial relationships, are the most influential factors affecting the participants' physiological well-being in this research.

We acknowledge that the true influence of each value is much more abstract than the assigned Shapley value due to the 'black box' nature of random forest regression; furthermore, we also acknowledge that model interpretability does not mean causality. Notwithstanding these caveats, we have chosen to share the Shapley plots in this paper because they allow us to examine the contributions of features in ways that can be easily understood and used to make inferences.

The associations reported in this paper bear out causal linkages reported by similar studies in the field. For example, Kessel et al. [38] documented positive relationships between ambient temperature and both body core temperature and corneal temperature, and Che Muhamed et al. [39] documented positive relationships between relative humidity and body core temperature.

Forte et al. documented that high cognitive functions are associated with higher heart rate variability [34], and Kazmi et al. documented that heart rate variability is inversely correlated to heart rate [35]; this reinforces our conclusion that the frontal and occipital lobes are most related to heart rate, which generally suggests that high cognitive functions lead to lower heart rate.

As for heart rate, Randall and Shelton found that "carbon dioxide excess causes an increase in ventilation volume by virtue of a greater depth of breathing, the frequency decreasing slightly. The heart rate goes up with increasing carbon dioxide concentrations" [40]. Verberkmoes et al. shared that "the influence of atmospheric pressure and temperature on the incidence of acute aortic dissections may be explained by an increase in sympathetic activity, which is responsible for higher blood pressure, and heart rate" [41]. Another factor that affects heart rate is dust concentration. As suggested in the research by Pope et al., elevated particulate levels were associated with increased mean heart rate and decreased overall heart rate variability [42]. Sound also plays a part in affecting heartbeat, according to past research—for example, that of Kraus et al. [43]—the higher the noise level, the higher the heart rate. Heart rate drops at night when humans are sleeping. According to Ahmed,

"a balance of impulse from the sympathetic and the parasympathetic nerves determine a person's baseline heart rate. Interestingly, in experiments where a person's nerve supply is blocked, the heart rate is often higher; this would suggest that the parasympathetic nerve impulses that serve to slow the heart rate down are the predominant force under normal resting conditions. This is particularly evident at night when most people have a significant drop in heart rate" [44].

In terms of the associative relationships between the microclimate and mental health, Mullins and White observed that "we find that higher temperatures increase emergency department visits for mental illness, suicides, and self-reported days of poor mental health" [45]. This position is congruent with that of the present study, in which extremes of sound levels were associated with both lower mental well-being, as indicated by the stress score [46].

As the present paper is inspired as a response to anthropogenic climate change, the extent to which carbon dioxide concentrations affect health and well-being is of interest [47]. Kajtár et al. have reported that well-being—as well as capacity to focus attention—both decline when carbon dioxide concentration in the air increases nearly tenfold to 3000 ppm [48].

The preceding parallels between earlier studies and the present study are encouraging, because a foundational paradigm driving our work was that of citizen science and the democratization of small-scale, low-cost research, as enabled by data science and the internet of things (IoT). We assembled the wearables from off-the-shelf parts and coded them ourselves. At the same time, DIY headsets were also designed and customized to each participant. Technology has enabled such democratization not only in terms of the (relatively) low cost of the hardware and the open-source movement in general, but also in terms of recent developments in data science and machine learning. The latter have meant that the large and burgeoning datasets associated with the use of IoT can be accessible and intelligible to wider cohorts of students and researchers. The parallels between our work and earlier studies suggest that—going forward, in a world where anthropogenic climate change is a (sad) reality—meaningful scientific and geographic inquiry can be undertaken by a much wider cross-section of the general public than was previously possible.

Several opportunities therefore suggest themselves for potential future work, which could take the form of either scaling up or translation to investigate other microclimatic variables (such as the role of infrared radiation on well-being) and socio-demographic contexts, such as, for example, investigating the productivity and attentiveness of students in the classroom settings, so as to make schools' classrooms a more conducive place for students to study [49]. These and other possible avenues for future work will go some way towards addressing the limitations of the present study, foremost among which was the movement restrictions associated with the necessity to follow the COVID-19 protocols throughout the duration of the study. These movement restrictions meant that the microclimates sampled were necessarily limited in variety. It is hoped that in the near future, when the prospect of another pandemic happening is lower, more collections of various data can be achieved. Furthermore, in the future, this can be our fundamental work towards greater understanding of neurological disorders and mental illnesses, as well as the unlocking of the keys for suitable treatment.

## 6. Conclusions

The study reported in this paper set out to investigate the associative relationships between microclimate, physiological responses, and brain activities. We approached the investigation from the perspective of citizen science, conceptualizing, designing, and fabricating what we could. We analyzed the resulting datasets informed by contemporary understandings of data science and machine learning.

Our results suggest that sound level, carbon dioxide concentration, and dust concentration feature more importantly in the regression models trained on our datasets. These findings are congruent with preceding studies, and we see a primary contribution of our work as the demonstration that—in an age of anthropogenic climate change—broader

cohorts of students, researchers, and the general public have potential access to tools, methods, and means of analyses that were once deemed only within the reach of a privileged few due to reasons of cost, fragility, and complexity.

It is our hope that our study contributes in a small way to a body of work to help urban planners, designers of living spaces, and caregivers—among many others—to conceptualize modes of human habitation in sympathy with the needs of our planet, in an era when humankind has more potential than it has ever had to influence biomes in general for better or for worse.

In conclusion, it is important to seek to understand the complex and non-linear relationships between the microclimate, physiological responses, well-being, and brain function. This paper has reported a study conducted by students in which wearable devices containing environmental sensors were designed and worn from December 2021. Over the same period, the data from these sensors were complemented by that obtained from smartwatches, and EEG data were also collected. We hope to contribute to the body of literature on the relationships between the microclimate, well-being, and brain activities. It is our hope that this wider body will catalyze subsequent studies to address the physical and mental health problems arising from climate change, in order to boost productivity and life satisfaction in the not-too-distant future.

**Author Contributions:** Conceptualization, K.Y.T.L., M.A.N.D. and M.T.N.T.; Data curation, M.A.N.D. and M.T.N.T.; Formal analysis, M.A.N.D. and M.T.N.T.; Investigation, M.A.N.D. and M.T.N.T.; Methodology, R.Y.; Project administration, K.Y.T.L.; Resources, K.Y.T.L.; Supervision, K.Y.T.L.; Validation, R.Y. and J.S.F.; Visualization, M.A.N.D. and M.T.N.T.; Writing—original draft, M.T.N.T.; Writing—review and editing, K.Y.T.L. All authors have read and agreed to the published version of the manuscript.

**Funding:** This research received no external funding.

**Institutional Review Board Statement:** The study was conducted in accordance with the Declaration of Helsinki and approved by the Ethics Committee of the National Institute of Education (protocol code CRPPIRB-2021-06-KL 2021).

**Informed Consent Statement:** Informed consent was obtained from all subjects involved in the study.

**Data Availability Statement:** The data presented in this study are available on request from the corresponding author.

**Acknowledgments:** We would like to acknowledge the following people: Ahmed Hazyl Hilmy, for his invaluable advice on building the prototype as well as resources; Tan Yan Li, for his invaluable guidance, support, and suggestions throughout the course of our research; Carlo Dalla Quercia, for his invaluable advice and help when we conducted our data analysis; and everyone who has helped us in one way or another, without whom we would not have been able to successfully complete our research.

**Conflicts of Interest:** The authors declare no conflict of interest.

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
