# Peer review of "Investigating the Effects of Microclimate on Physiological Stress and Brain Function with Data Science and Wearables"

_sustainability, doi:10.3390/su141710769_

Round 1
Reviewer 1 Report
The manuscript of Lim et al. reported the influence of microclimate on physiological stress and brain functions, using the result of data science and wearable devices. The research is interesting and contributive. The manuscript has been well-written.
1. The authors listed 9 keywords. Some of them could be removed.
2. The authors are required to enlarge the part 2.
3. Were the environmental data and EEG signals tested during the motion or resting state of the human body? How do the authors ensure the accuracy of the sensors during the moving process?
4. How to ensure the accuracy of the EEG data tested from the DIY headset?
5. The influence of noise is not mentioned in the part 1 and part 2. The authors are required to explain the reason to use sound as one of the environmental variables.
6. Some of the figures are totally indiscernible.
Author Response
Response to reviewers
We would like to express our sincere appreciation to the reviewers for investing their time in shaping this manuscript for the better.
We have taken care to understand the intent behind the suggestions for improvement, and to address each of these to the best of our ability.
Reviewer 1
- The authors listed 9 keywords. Some of them could be removed
We have verified with the Guidance to authors provided on the journal’s website that ten keywords are permissible. We would please seek to retain the existing nine keywords. Thank you.
https://www.mdpi.com/journal/sustainability/instructions/
- The authors are required to enlarge the part 2
Thank you. We have added additional literature to this section with respect to EEG signals through seven additional bibliographic references.
Beaty, R. E., Benedek, M., Wilkins, R. W., Jauk, E., Fink, A., Silvia, P. J., ... & Neubauer, A. C. (2014). Creativity and the default network: A functional connectivity analysis of the creative brain at rest. Neuropsychologia, 64, 92-98.
Dmochowski, J. P., Sajda, P., Dias, J., & Parra, L. C. (2012). Correlated components of ongoing EEG point to emotionally laden attention–a possible marker of engagement?. Frontiers in human neuroscience, 6, 112.
Dmochowski, J. P., Bezdek, M. A., Abelson, B. P., Johnson, J. S., Schumacher, E. H., & Parra, L. C. (2014). Audience preferences are predicted by temporal reliability of neural processing. Nature communications, 5(1), 1-9.
Ki, J. J., Kelly, S. P., & Parra, L. C. (2016). Attention strongly modulates reliability of neural responses to naturalistic narrative stimuli. Journal of Neuroscience, 36(10), 3092-3101.
Krishnan, P., & Yaacob, S. (2019). Drowsiness detection using band power and log energy entropy features based on EEG signals. Int. J. Innov. Technol. Explor. Eng, 8, 830-836.
Murata, A., Uetake, A., & Takasawa, Y. (2005). Evaluation of mental fatigue using feature parameter extracted from event-related potential. International journal of industrial ergonomics, 35(8), 761-770.
Paszkiel, S. (2017, March). Characteristics of question of blind source separation using Moore-Penrose pseudoinversion for reconstruction of EEG signal. In International Conference Automation (pp. 393-400). Springer, Cham.
- Were the environmental data and EEG signals tested during the motion or resting state of the human body? How do the authors ensure the accuracy of the sensors during the moving process
Thank you. We have clarified in the description of the validation procedure that participants were in resting state seated individually in respective rooms.
- How to ensure the accuracy of the EEG data tested from the DIY headset
Care was taken to ensure the accuracy of the data from the DIY headset by requiring participants to go through two sessions, first with a DIY EEG headset and then with an industrial grade ANT Neuro EEG headset, according to the protocol described in the procedure section (see Figure 4, for example). Figure 5 illustrates the relative differences in the data drawn from the two kinds of headsets, with Table 1 verifying that the data is comparable between the two datasets.
- The influence of noise is not mentioned in the part 1 and part 2. The authors are required to explain the reason to use sound as one of the environmental variables
Thank you. In the second paragraph of the Literature Review, we include a bibliographic reference (Jafari et al., 2019) to the importance of noise as an environmental stressor. Specifically, “with the rise in noise levels, the relative power of the Alpha band increases while the relative power of the Beta band decreases as compared to background noise. The most prominent change in the relative power of the Alpha and Beta bands occurs in the occipital and frontal regions of the brain respectively.”
- Some of the figures are totally indiscernible
We apologise for this oversight and have taken care to replace the illegible figures.
Reviewer 2 Report
Dear authors
I would like to thank you for giving me the opportunity to review the manuscript entitled “Investigating the effects of microclimate on physiological stress and brain function with data science and wearables”. I really enjoyed reading this article. The aim of this study was to investigate the effects of microclimate on physiological stress and brain function sing data science. This work is high quality and provide valuable information regarding the effects of microclimate on physiological stress and brain function using deep learning. I have only few minor comments. My comments are as follows:
- The aim of the study should be provided in the Abstract.
- Line 72-80: Please move this paragraph to the Methods section.
- Line 81: Please remove the year (2020) because stating the year is for APA referencing style.
- In one paragraph at the end of the Introduction specify the aims and hypotheses of the study.
Author Response
Response to reviewers
We would like to express our sincere appreciation to the reviewers for investing their time in shaping this manuscript for the better.
We have taken care to understand the intent behind the suggestions for improvement, and to address each of these to the best of our ability.
Reviewer 2
- The aim of the study should be provided in the Abstract.
Thank you for pointing out our blindspot. We have amended the Abstract accordingly.
- Line 72-80: Please move this paragraph to the Methods section.
We have re-positioned the paragraph and concur this improves the readability and structure of the manuscript.
- Line 81: Please remove the year (2020) because stating the year is for APA referencing style.
This has been done, thank you.
- In one paragraph at the end of the Introduction specify the aims and hypotheses of the study.
This has also been written, thank you.